# Energy transfer-mediated multiphoton synergistic excitation for selective C($sp^3$)−H functionalization with coordination polymer

Zhonghe Wang[1,3], Yang Tang[1,3], Songtao Liu[1,3], Liang Zhao [1] ✉, Huaqing Li[1], Cheng He[1] & Chunying Duan [1,2] ✉

Activation and selective oxidation of inert C($sp^3$)−H bonds remain one of the most challenging tasks in current synthetic chemistry due to the inherent inertness of C($sp^3$)−H bonds. In this study, inspired by natural mono-oxygenases, we developed a coordination polymer with naphthalenediimide (NDI)-based ligands and binuclear iron nodes. The mixed-valence $Fe^{III}Fe^{II}$ species and chlorine radicals ($Cl^{\bullet}$) are generated via ligand-to-metal charge transfer (LMCT) between $Fe^{III}$ and chlorine ions. These $Cl^{\bullet}$ radicals abstract a hydrogen atom from the inert C($sp^3$)−H bond of alkanes via hydrogen atom transfer (HAT). In addition, NDI converts oxygen to $^1O_2$ via energy transfer (EnT), which then coordinates to $Fe^{II}$, forming an $Fe^{IV}=O$ intermediate for the selective oxidation of C($sp^3$)−H bonds. This synthetic platform, which combines photoinduced EnT, LMCT and HAT, provides a EnT-mediated parallel multiphoton excitation strategy with kinetic synergy effect for selective C($sp^3$)−H oxidation under mild conditions and a blueprint for designing coordination polymer-based photocatalysts for C($sp^3$)−H bond oxidation.

Developing a high-performance catalytic approach for the oxidation of C($sp^3$)−H bonds to value-added fine chemicals affords economic and ecological benefits in modern chemistry[1–4]. Natural oxygenases with sterically confined spaces and appropriately positioned functional groups can oxygenate different kinds of inert C−H bonds with high substrate fidelity[5–7]. These oxygenases can be considered a blueprint for developing metal catalysis. However, getting the same efficiency and selectivity as natural oxygenases in an artificial system is still challenging[8,9]. Adopting the similar structure and function of natural oxygenases in a metal catalyst for C−H bond oxidation with the aid of oxygen activation has proven to be an effective strategy[10]. Previously, our group and other researchers developed a consecutive multi-photon excitation strategy, which targeted to activate inert C($sp^3$)−H bonds, employing photoinduced electron transfer (PET), ligand-to-metal charge transfer (LMCT), and hydrogen atom transfer (HAT) in an integrated system[11–14]. Nevertheless, the intrinsic instability and shorter

lifetime of the transient state and its corresponding excited state impede the efficacy of the second excitation; consequently, the active radical intermediate and inert substrate participate in HAT under the typical paradigm of diffusion-limited steps[15,16]. Hence, promoting the efficacy of consecutive photoexcitation by simultaneously achieving a highly stable intermediate state and sufficient thermodynamic driving force for the excited state is highly desirable for photocatalytic transformations.

Dye-loaded coordination polymers could efficiently activate oxygen and control the selectivity of the oxygenation product via photoinduced energy transfer (EnT)[17,18]. We previously combined light-driven EnT to simultaneously activate oxygen into active singlet oxygen ($^1O_2$) species and activate the C($sp^3$)−H bond via a photomediated LMCT and HAT; this strategy achieves high reactivity and selectivity for the C($sp^3$)−H bond oxidation in a heterogeneous manifold via the fixation and dispersion of highly active sites similar to enzymes into

[1]State Key Laboratory of Fine Chemicals, Frontier Science Center for Smart Materials, School of Chemical Engineering, Dalian University of Technology, Dalian 116024, People's Republic of China. [2]State Key Laboratory of Coordination Chemistry, Nanjing University, Nanjing 210093, People's Republic of China. [3]These authors contributed equally: Zhonghe Wang, Yang Tang, Songtao Liu. ✉e-mail: zhaol@dlut.edu.cn; cyduan@dlut.edu.cn

coordination polymers[19]. The two photons excitation with separate photon absorption at both active sites would avoid the photoexcitation of the in situ-formed transient state in the consecutive multiphoton excitation and two-photon excitation absorption, expanding the accessibility of thermodynamically demanding reactions.

Thinking outside the box of coordination polymers could afford a fine-tunable catalytic platform such as the integration of dye-based bridging ligands and functionalized metal nodes into one network, thus realizing multiple synergistic catalytic processes in tandem[20,21]. In this study, inspired by the reactivity and selectivity of iron-based oxygenases, we developed a coordination polymer-based photocatalyst using a powerful crystal engineering manifold[22,23]. Abundant iron ions and a commercially available used organic dye, naphthalenediimide (NDI), were incorporated into a redox-active coordination polymer to combine photoinduced EnT, LMCT, and HAT by an EnT-mediated parallel multiphoton excitation strategy with kinetic synergy effect (Fig. 1a).

These binuclear Fe[III] species, which are uniformly distributed on the surface of the coordination polymers, can capture chlorine ions, achieving an LMCT excited state to generate chlorine radical and the mixed-valence binuclear Fe[III]Fe[II] under light irradiation[24]. The in situ-formed chlorine radical would directly abstract a hydrogen atom from an inert C(sp³)–H bond, generating an alkyl radical, even for the extremely inert methane C–H bond via HAT[25]. Parallel to this LMCT excitation and HAT, the NDI groups would absorb another photon to reach the excited state, promoting the activation of $O_2$ into $^1O_2$ via EnT[26]. Then, the mixed valence Fe[III]Fe[II] species would interact with the in situ-formed $^1O_2$, affording Fe-OOH species, the O–O bond in Fe-OOH breaks to afford the Fe[IV]=O species for oxidation transformation[27,28]. These catalysts would exhibit an outstanding alcohol/ketone selectivity and could be recycled for multiple rounds of

catalysis. The intensity-dependent experiments demonstrated a linear correlation between the yield and the quadratic of the photon power, corroborating the multiphoton nature of the catalytic reaction[11,16,29]. This is an example of a coordination polymer that combines photoinduced EnT and LMCT and provides a parallel excitation strategy with kinetic synergy effect for C(sp³)–H bond activation and selective oxidation under mild conditions. The well-defined structural characteristics and finely modified catalytic properties provide opportunities for in-depth mechanistic studies and serve as a blueprint for developing environmentally benign routes for the direct activation and oxidation of inert alkanes.

## Results

### Preparation and characterization of Fe–NDI

The solvothermal reaction of FeCl₃·6H₂O and the ligand N,N′-bis(5-isophthalic acid)naphthalenediimide (H₄BINDI) in N,N′-dimethylformamide (DMF, 5.0 mL) containing acetic acid (0.1 mL) and H₂O (0.1 mL) at 120 °C for 3 days produced the NDI-containing binuclear iron-based coordination polymer (Fe–NDI) in 35% yield. Single-crystal analysis revealed that Fe–NDI has a three-dimensional (3D) structure in the hexagonal space group $P2_1/c$ (Fig. S1). Each Fe ion is six-coordinated in an octahedral geometry and coordinates with two DMF molecules, in addition to four oxygen atoms from three carboxylic groups of different ligands (Fig. 1b). This suggests that Fe–NDI could have unsaturated coordinated metal sites for catalysis[30]. Two congruent Fe[III] ions are interconnected through carboxylate moieties derived from a pair of distinct ligands, forming distinctive secondary building units that serve as quartet connected nodes for the two-dimensional (2D) framework. Meanwhile, each ligand itself also functions as a quartet connected node, bridging four separate binuclear Fe₂ clusters via bidentate carboxylate moieties, culminating in a 2D

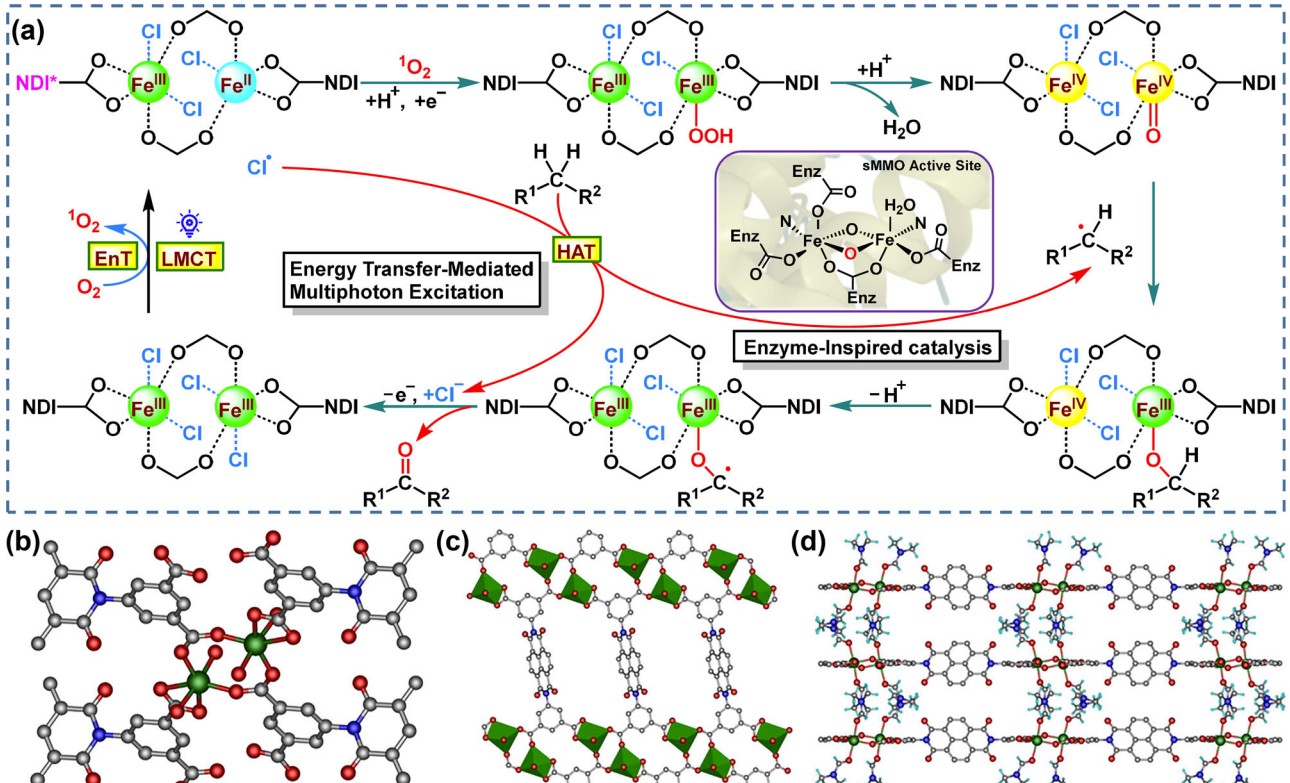

**Fig. 1 | Schematic of EnT-mediated multiphoton synergistic excitation strategy and crystal structure of Fe–NDI. a** The assumed photocatalytic C(sp³)–H bond activation and oxidation mechanism with binuclear iron-based coordination polymer. **b** Coordination structure of the binuclear Fe[III]Fe[III] for Fe–NDI. **c** Parallelogram channels inside the Fe–NDI layers. **d** Hydrogen bonding between the Fe–NDI layers. Color codes: Fe, green; C, gray; O, red; N, blue; H, cyan.

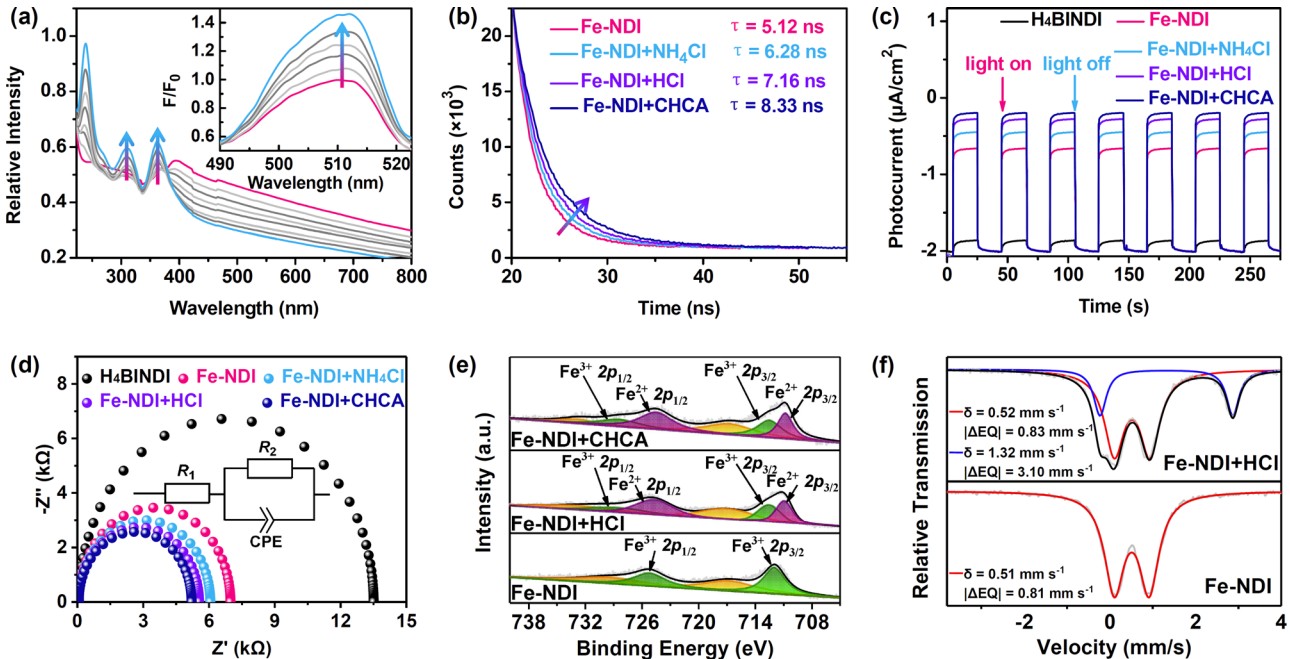

**Fig. 2 | Characterization of Fe−NDI. a** UV−vis spectra of Fe−NDI in HCl. Inset: Fluorescence spectra of Fe−NDI in HCl. **b** Time-dependent luminescence decay spectra of Fe−NDI and Fe−NDI soaked in HCl, NH₄Cl and cyclohexanecarboxylic acid (CHCA) solutions. **c** Photocurrent curves of H₄BINDI, Fe−NDI and Fe−NDI soaked in HCl, NH₄Cl and CHCA solutions with 455 nm LED irradiation. **d** Impedance curves of H₄BINDI, Fe−NDI and Fe−NDI soaked in HCl, NH₄Cl and CHCA solutions (R resistor, CPE constant phase element). **e** X-ray photoelectron (XPS) spectra of Fe−NDI and Fe−NDI soaked in HCl and CHCA solutions with 455 nm LED irradiation. **f** Mössbauer spectrum of Fe−NDI and Fe−NDI soaked in HCl with 455 nm LED irradiation.

plane formation. (Figs. 1c and S2). Moreover, the DMF molecules are coordinated to the iron ions along the *c*-axis of Fe−NDI. These 2D layers are further assembled into a 3D coordination polymer with an inter-lamellar distance of ~7.50 Å via multiple hydrogen bonds (3.20 Å) between the DMF molecules in adjacent layers (Figs. 1d and S3).

Notably, the binuclear Fe^III secondary building units in this Fe−NDI is formed by interacting with carboxylic groups to generate a separation of ~4.4 Å, which is similar to that of the natural binuclear iron monooxygenase active site (Fig. S4)[31]. The formation of such a binu-clear Fe₂ model helps to directly activate oxygen, boosting the oxida-tion of the inert C−H bonds. Moreover, the facile movement of coordinated DMF molecules creates vacant coordination points, which are ideal for the combination with radical precursors, such as chlorine ions and alkoxide. This is conducive to promoting the generation of highly electrophilic radicals through a light-mediated LMCT event, which will facilitate the activation of inert C−H bonds, encompassing those in methane and other light alkanes[25,32].

The phase purity of the bulk specimens was unveiled through elemental analysis and powder X-ray diffraction (Fig. S5). Elemental mapping images demonstrate a uniform distribution of all elements within the Fe−NDI framework with an iron content approximately 3.74 wt% (Fig. S6). Thermogravimetric analysis (TGA) showed the expulsion of solvent molecules from crystalline Fe−NDI in the tem-perature range of 50−300 °C, with the onset of Fe−NDI structural decomposition observed around 420 °C, confirming its structural integrity at room temperature (catalytic reaction condition) (Fig. S7). The DMF molecules coordinated to iron can be removed through supercritical fluid extraction[33]. The IR spectrum of Fe−NDI after supercritical fluid extraction shows that the characteristic vibration peak of the carbonyl group of DMF disappeared after treatment. Moreover, the IR spectrum of Fe−NDI soaked in a CH₃CN solution containing HCl (0.05 M) for 12 hours showed no apparent variation (Fig. S8). When Fe−NDI was soaked in a mixture solution containing an aqueous HCl solution with pH 2.0−7.0 and CH₃CN for 12 h[34], more than

90% of the Fe−NDI was recovered (Fig. S9), indicates that Fe−NDI possesses ideal stability in acidic solutions, which confirms its suit-ability for heterogeneous catalysis[35].

The solid-state ultraviolet−visible (UV−vis) spectra of Fe−NDI showed an absorption peak at 370 nm corresponding to the ligand H₄BINDI. This peak shifted to the visible region in Fe−NDI compared with that of the pure ligand H₄BINDI (Figure S10), reflecting its favor-able visible-light harvesting ability[36]. The addition of HCl (25 mM) to the CH₃CN suspension of Fe−NDI resulted in the appearance of sig-nificant UV−vis absorption peaks at 310 and 360 nm, which can be attributed to the LMCT absorption peaks of the Fe−Cl chromophore[37,38], whereas the decreasing UV−vis absorption band at >375 nm might be attributed to the dilution effect of the added solu-tion. At the same time, after filtering the heterogeneous Fe−NDI, the UV−vis spectra of the filtrate showed the absence of absorption peaks at 310 and 360 nm and the ICP-MS test of the filtrate showed no free iron ions. In addition, the fluorescence emission peaks of Fe−NDI suspensions were significantly increased at 455 nm excitation after the addition of HCl (Figs. 2a and S11−S12). These results suggest that Fe−NDI can combine with chlorine ions by coordinating with the Fe nodes[25]. Furthermore, the emission lifetime increased from 5.12 to 7.16 ns (Fig. 2b), which is in agreement with the fluorescence titration results. Hence, the coordination with chlorine ions increases the emission lifetime of Fe−NDI and affords more opportunities for the activation of the inert C−H bonds through photoexcited LMCT to produce highly electrophilic chlorine radicals. The transient photo-current responses of Fe−NDI showed a reproducible photocurrent upon on/off cycles of 455 nm LED irradiation, which is in contrast to the faint photocurrent response shown by the alone ligand H₄BINDI (Fig. 2c). This suggests that Fe−NDI demonstrates excellent electron-hole separation[39]. Electrochemical impedance spectroscopy (EIS) measurements showed that Fe−NDI exhibits a lower electron transport resistance ($R_{ct}$) than that of H₄BINDI (Fig. 2d), demonstrating that Fe−NDI rapidly transfers electrons to ensure high catalytic efficiency[40].

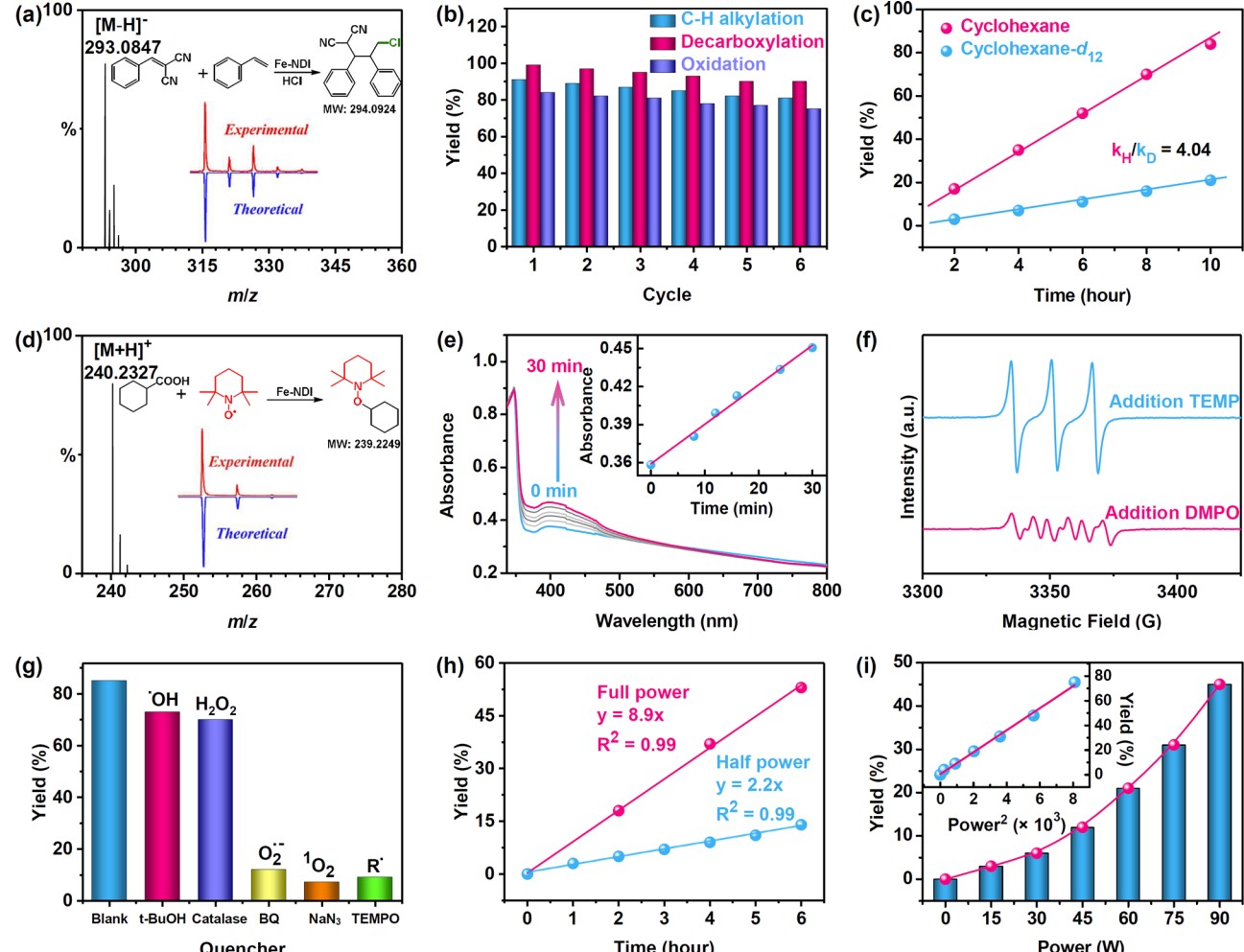

**Fig. 3 | Photocatalysis properties of Fe−NDI. a** Chlorine radical trapping experiment in direct C($sp^3$)−H alkylation using styrene as the trapping agent. **b** Recycled experiments for alkylation, decarboxylation and oxidation. **c** Kinetic isotope effect (KIE) experiment. **d** Cyclohexyl radical trapping experiment in decarboxylative functionalization using 2,2,6,6-tetramethylpiperidinooxy (TEMPO) as the trapping agent. **e** Time-dependent UV−vis absorption spectra of 3,3′,5,5′-tetramethylbenzidine (TMB) oxidation by Fe−NDI. **f** EPR spectra for the oxidation reaction. **g** Relationship between the yield of oxidative reaction and additives. **h** Relationship between yield and full power or half power of light. **i** Linear relationship between quadratic light intensity and yield in the oxidation reaction.

## Photocatalytic C($sp^3$)−H bond alkylation

Soaking Fe−NDI in $CH_3CN$ containing HCl (0.05 M) increased the photocurrent further and decreased the resistance (Fig. 2c, d). Hence, the coordination of chlorine ions improved the electron transfer efficiency, which was conducive to providing Cl· radicals via LMCT under photoexcitation. The chemical state of Fe ions in Fe−NDI was examined via X-ray photoelectron spectroscopy (XPS). The Fe $2p_{3/2}$ and Fe $2p_{1/2}$ binding energy peaks at 711.9 and 725.2 eV, respectively, verified the presence of the $Fe^{III}$ state (Fig. 2e)[41]. The zero-field $^{57}$Fe Mössbauer spectrum of Fe−NDI tests was executed at 80 K. The result showed a symmetric doublet with isomer shift $\delta = 0.51$ mm s$^{-1}$ and quadrupole splitting $|\Delta EQ| = 0.81$ mm s$^{-1}$, which was attributed to the high-spin Fe(III) species in Fe−NDI (Fig. 2f)[42]. Irradiation of Fe−NDI containing HCl by an 455 nm LED led to the appearance of peaks at 710.8 and 724.6 eV corresponding to $Fe^{II}$ (Fig. 2e)[41]. And the zero-field $^{57}$Fe Mössbauer spectrum showed a mixed peak pattern. Through data fitting, the red peak pattern showed isomer shift $\delta = 0.52$ mm s$^{-1}$ and quadrupole splitting $|\Delta EQ| = 0.83$ mm s$^{-1}$, which was attributed to a high-spin Fe(III) iron species. The blue peak pattern showed isomer shift $\delta = 1.32$ mm s$^{-1}$ and quadrupole splitting $|\Delta EQ| = 3.10$ mm s$^{-1}$, which was attributed to a high-spin Fe(II) iron species[43]. A 27% conversion from the high-spin Fe(III) species to high-spin Fe(II) species in Fe−NDI under the irradiation of a 455 nm LED verifies the occurrence of the LMCT process (Fig. 2f). Moreover, electrospray ionization mass spectrometry (ESI-MS) for the reaction of 2-benzylidenemalononitrile (**1**), styrene, HCl and Fe−NDI in $CH_3CN$ solution at room temperature for 12 h under a 455 nm LED irradiation captured the adduct of the Cl· radical and alkenes (Figs. 3a and S13). Meanwhile, Ts protected bis-allyl amine as another chloro acceptor was added into alkylation reaction catalyzed by Fe−NDI. Both $C_{13}H_{18}ClNO_2S$ and $C_{13}H_{17}Cl_2NO_2S$ could be detected from ESI-MS (Fig. S14)[44], further indicating the formation of chlorine radicals. As Cl· radicals are only formed from the homolysis of $Fe^{III}$−Cl, the chlorine ions should have been coordinated to Fe−NDI. This suggests that the in situ-formed $Fe^{III}$−Cl chromophore would absorb photons to trigger LMCT from the chlorine anions to the $Fe^{III}$ node of Fe−NDI to form $Fe^{II}$ and Cl· radicals for further activation of C($sp^3$)−H bonds via HAT[24].

Subsequently, we investigated the photocatalytic performance of Fe−NDI in the activation of inert C($sp^3$)−H bonds. The irradiation of a $CH_3CN$ solution (1.0 mL) of Fe−NDI (5.0 μmol), **1** (0.1 mmol), cyclohexane (1.0 mmol) and HCl (0.05 mmol) with a 455 nm LED at room temperature in argon for 12 h produced a 91% yield of the desired product (Table S4, entry 1). However, the reaction did not occur in the absence of Fe−NDI, HCl, or light, suggesting their indispensability for the reaction (Table S4, entries 2−4). Moreover, increasing the HCl concentration from 0.01 to 0.05 mmol increased the rate of alkylation,

which could be attributed to the increased concentration of Cl· radicals for the activation of C($sp^3$)−H bond during the alkylation reaction (Fig. S15). The dynamic tracing experiment results showed that the reaction yield increased rapidly as the reaction time increased to a maximum value within 12 h (Fig. S16). In addition, Fe−NDI was filtered out of the reaction system after 4 h of the alkylation reaction, and then the filtrate was kept in the light until 14 h (Fig. S17). The results showed that the yields did not change after the Fe−NDI was filtered out. Moreover, we also performed filtration experiments in parallel after 6 and 8 h, respectively, and the yields did not improve further. While the ICP−MS testing of the filtrate revealed that Fe ions were not detected in the solution. Those results indicated that Fe−NDI effectively inherited the ability of Fe−Cl bonds to activate C−H bonds and had good stability during the catalytic process, without forming a homogeneous FeCl₃ catalytic system[45]. Next, we conducted catalyst recycling experiments to investigate the recyclability of Fe−NDI. After each round of the reaction, the solution was removed and replaced with a fresh solution harboring the correlative substrates for the next round. After six consecutive cycles, there was no obvious decline in yield (Fig. 3b), highlighted that the Fe−NDI possessed the well chemical stability and remarkable catalytic activity[46].

Furthermore, we investigated the substrate scope of the C($sp^3$)−H bond activation reactions in the presence of Fe−NDI under the optimized conditions. The reactions of cycloalkanes and **1** produced excellent yields (Fig. 4, **3a**−c). Cyclic ether and ester selectively afforded the corresponding alkylation products in good-to-excellent yields (Fig. 4, **3d**−**g**). Alkyl ethers and esters exhibited selective C−H alkylation at the most electron-rich sites of the C($sp^3$)−H nucleophiles. In addition, the C($sp^3$)−H bond of toluene furnished the desired product in good yield (Fig. 4, **3h**). Moreover, high regioselectivity was observed in all cases. Furthermore, we applied Fe−NDI for the radical alkylation reactions of methane and other gaseous alkanes in the form of alkylating feedstock. As shown in Fig. 4, methylation of **1** using methane afforded the desired product in 21% yield (Fig. 4, **3i**). Other gaseous alkanes were activated at normal temperatures and pressures in moderate-to-excellent yields (Fig. 4, **3j**−**l**).

### Proposed mechanisms of catalysis
From a mechanistic point of view, the photoexcitation of high-density distributed Fe$^{III}$−Cl chromophores on Fe−NDI generates Cl· radicals and Fe$^{II}$ through LMCT under one-photon irradiation (Fig. S18)[24]. Cl· radicals further abstract a hydrogen atom from the C($sp^3$)−H bond to generate highly reactive alkyl radicals via typical HAT for substrate transformations[47]. The alkyl radicals are captured by the electron-deficient olefin to form radical intermediates, which undergo single electron transfer (SET) with the in situ-formed Fe$^{II}$ to generate coupling products and regenerate Fe$^{III}$. TEMPO completely inhibits the formation of the product, thus confirming the participation of radicals in the reaction mechanism[48]. Moreover, the cyclohexyl radical−TEMPO adduct was detected by ESI-MS, which confirms C($sp^3$)−H bond activation via radical formation (Fig. S19). To gain further insight into the reaction mechanism, deuterium labeling experiments were conducted, where **1** was treated with cyclohexane ($k_H$) and cyclohexane-$d_{12}$ ($k_D$) in two different vessels under standard conditions. A kinetic isotope effect value ($k_H/k_D$) of 4.04 was observed for the alkylation of the C−H bonds in cyclohexane (Fig. 3c). This result indicates that C−H bond cleavage is the rate-determining step (RDS) in the C−H alkylation process[49,50].

### Photocatalytic decarboxylative functionalization
To further explore the capacity of Fe−NDI to generate carbon radical via LMCT, it was applied to the decarboxylative functionalization of aliphatic acids[51]. The lifetime, photocurrent, impedance, and XPS results of Fe−NDI upon the addition of cyclohexanecarboxylic acid (CHCA) were in agreement with those of HCl (Figs. 2b−e and S20−S21),

indicating that the irradiation of Fe−NDI triggered LMCT from the coordinated CHCA to the Fe$^{III}$ node of Fe−NDI to form Fe$^{II}$, accompanied by the formation of a cyclohexyl radical. Moreover, the reaction of CHCA, 2,2,6,6-tetramethylpiperidinooxy (TEMPO) and Fe−NDI in CH₃CN solution under irradiation of 455 nm LED in argon at room temperature for 6 h produced cyclohexyl radicals, which were detected by ESI-MS (Figs. 3d and S22)[48,52]. Irradiation of a 1,4-dioxane solution (1.0 mL) of Fe−NDI (5.0 μmol), **1** (0.1 mmol) and CHCA (0.5 mmol) under irradiation of 455 nm LED for 12 h at room temperature in argon produced the desired product in 99% yield (Fig. 4, **5a**). The yield was also maintained after six cycles toward Fe−NDI (Fig. 3b). Various aliphatic acids, including primary and secondary carboxylic acids, delivered excellent yields of the corresponding products (Fig. 4, **5b**−**i**). 1-Adamantane-acetic acid with a rigid skeleton afforded the product in 69% yield (Fig. 4, **5j**) and 4-methoxyphenylacetic acid and phenylglyoxylic acid furnished the product in 78% and 76% yields, respectively (Fig. 4, **5k**, **l**). Pharmaceutical molecules with carboxylic acids, such as gemfibrozil capsules and ibuprofen, furnished moderate yields, indicating that lamellar Fe−NDI was compatible with large bulking substrates (Fig. 4, **5m**, **n**).

### Photocatalytic C($sp^3$)−H bond oxidation
NDI-decorated Fe−NDI demonstrates a reproducible photocurrent response and low electron transport resistance, which facilitate the generation of reactive oxygen species (ROS) to oxidize substrates for tandem conversion. Therefore, the 3,3′,5,5′-tetramethylbenzidine (TMB) oxidation experiments were performed to verify ROS production from Fe−NDI[53]. The addition of Fe−NDI to a CH₃CN solution of TMB progressively increased the absorption peak intensity at 397 nm with increasing irradiation time (Fig. 3e), indicating the generation of ROS. To further clarify the type of ROS generated in the presence of Fe−NDI, EPR spectra were recorded in the presence of 2,2,6,6-tetramethylpiperidine (TEMP) and 5,5-dimethyl-1-pyrroline N-oxide (DMPO) as ROS trapping agents (Fig. 3f)[54−56]. Both $^1O_2$ and the superoxide radical $O_2^{·−}$ were observed. Furthermore, oxidation of α-terpinene over Fe−NDI was performed under an oxygen atmosphere; the catalytic selectivity toward ascaridole and p-cymene were 65% and 13%, respectively, suggesting that $^1O_2$ was the major ROS[57].

Subsequently, the selective oxidation of inert C($sp^3$)−H bonds was selected as the benchmark for investigating the performance of Fe−NDI in the synergistic activation of C($sp^3$)−H bonds and oxygen through two parallel photoexcitation. In a typical reaction condition, irradiation of a CH₃CN solution of Fe−NDI (5.0 μmol), cyclohexane (0.2 mmol) and NH₄Cl (0.05 mmol) under LED light (455 nm) within 12 hours at room temperature in air exhibited an 85% conversion with a perfect selectivity of >95% toward cyclohexanone (Fig. 5, **6a**). No significant decrease in yield was observed and the selectivity was always maintained above 95% after six cycles toward Fe−NDI (Fig. 3b), displaying high stability and catalytic efficiency of Fe−NDI[46,58].

For the other inert alkane, the oxidation system could also provide the corresponding ketone with pleasing yields. Cyclopentane, cycloheptane and cyclooctane all held for that oxidation reaction, delivering satisfactory yields (Fig. 5, **6b**−**d**, 78−89%). Apart from alkane, ether was compatible with that oxidation reaction and yielded the corresponding ester or lactone with gratifying yields (Fig. 5, **6e**−**l**, 65−81%). It was worth mentioning that those substrates with the benzyl group had high activity in the oxidation reaction since the benzyl radical was more stable compared to the above-mentioned alkyl radicals (Fig. 5, **6m**−**p**, 85−95%). Whereas, toluene was oxidized to benzaldehyde (10%) and benzoic acid (63%), which was due to the deep oxidation of the generated benzaldehyde to benzoic acid catalyzed by Fe−NDI. Therefore, the combination of the Cl· radicals formed via LMCT of Fe−NDI and binuclear iron nodes in Fe−NDI that resemble the active centers of natural monooxygenases could facilitate the direct activation and oxidation of inert alkanes

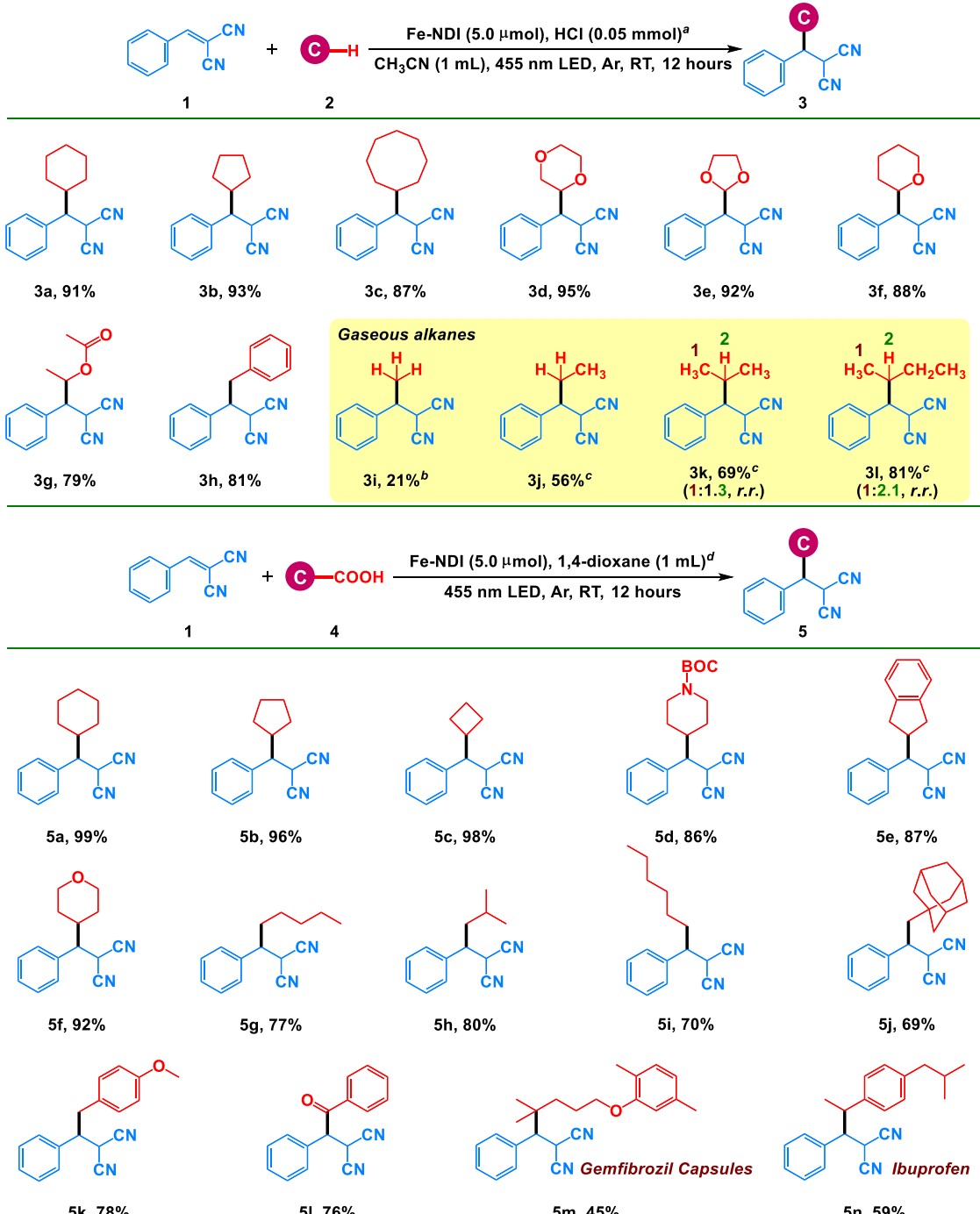

**Fig. 4 | Substrate scope of alkylation and decarboxylative functionalization.** a Standard Conditions: **1** (0.1 mmol), cyclohexane (1.0 mmol), Fe–NDI (5.0 μmol) and HCl (0.05 mmol, concentrated HCl) in CH₃CN (1 mL) for 12 h at room temperature with 455 nm LED; **b 1** (0.1 mmol), methane (5 MPa), Fe–NDI (15.0 μmol) and HCl (0.1 mmol, concentrated HCl) in CH₃CN (7 mL) for 48 h; **c 1** (0.2 mmol), balloon with light alkane, Fe–NDI (10.0 μmol) and HCl (0.1 mmol, concentrated HCl) in CH₃CN (7 mL) for 48 h. **d** Standard Conditions: **1** (0.1 mmol), **4** (0.5 mmol), Fe–NDI (5.0 μmol) in 1,4-dioxane (1 mL) for 12 h.

under environmentally benign conditions using oxygen as the oxidant. While photoexcited $FeCl_3$ exhibits a commendable activity in the activation of C($sp^3$)–H bonds, chemists still remains a great challenge to use molecular oxygen to directly and selectively oxidize the inert C($sp^3$)–H bonds in alkanes into more complex value-added chemicals under mild conditions[59–61]. To date, scarce literatures exist to show that $FeCl_3$ can facilitate the oxidation of inert C($sp^3$)–H bonds[62,63], however, the outcomes have been modest in terms of yields and selectivity because the low selectivity in ROS production

under ultraviolet light and the low activity of homogeneous $FeCl_3$ for boosting O–O bond cleavage.

To investigate the mechanism on the oxidation of C($sp^3$)–H bonds, the same oxidative reaction was performed with different ROS quenchers to further explore the role of each ROS (Fig. 3g)[64]. The hydroxyl radical (·OH) scavenger, $t$-BuOH, slightly decreased the product yield, suggesting that ·OH might be involved in this process but is not the major ROS controlling oxidation[65]. While the oxidation yield was maintained 83% of the initial value with the addition of catalase

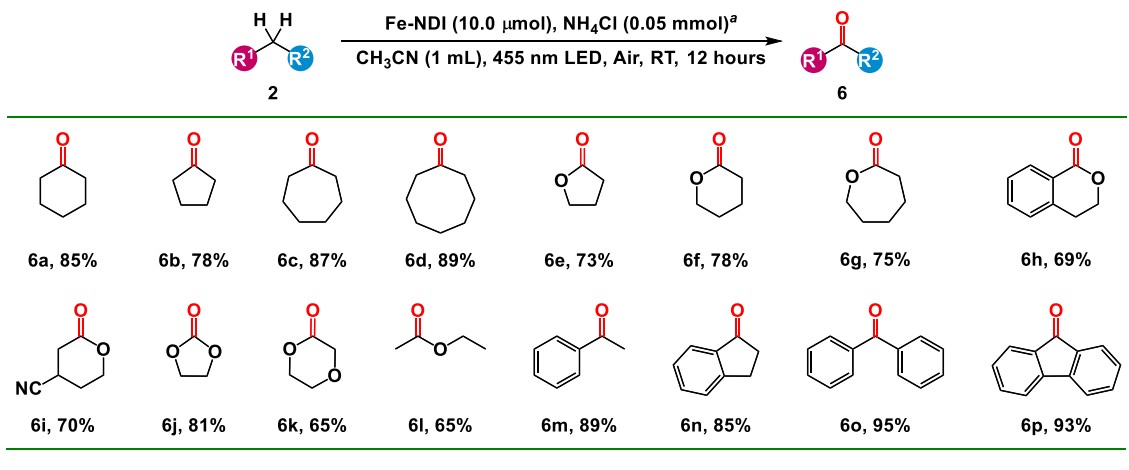

**Fig. 5 | Substrate scope of C–H activation and oxidation. a** The reaction was performed with 0.2 mmol alkane, 5 mol% Fe–NDI, 0.05 mmol NH₄Cl in 1 mL CH₃CN at room temperature for 12 h under air.

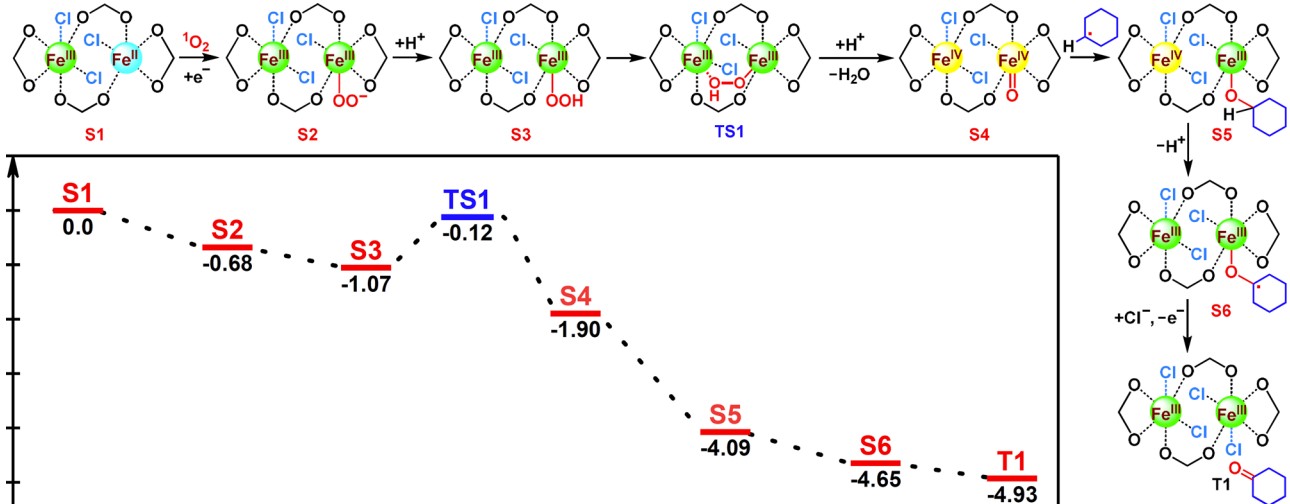

**Fig. 6 | DFT calculations.** Schematic of the proposed C–H activation and oxidation pathway of cyclohexane on Fe–NDI and the Gibbs free energy calculations of each intermediate in the pathway.

(CAT), the specific quenching agent of $H_2O_2$, indicating that $H_2O_2$ was not a major contributor to the oxidation reaction[66]. The NaN₃ and the *p*-benzoquinone significantly decreased the yields to 12% and 8%[67], respectively, suggesting that $^1O_2$ and $O_2^{•-}$ play a significant role in the oxidative process. Considering the results of the oxidation of α-terpinene, we think $O_2^{•-}$ was likely derived from the combination of $^1O_2$ and the $Fe^{II}$ species of Fe–NDI[68]. The radical capturing agent TEMPO significantly decreased the yield of the target product to only 9%, proving that the reaction involves the generation of free radicals (Fig. 3g)[48].

To understand the interaction of binuclear iron assisted ROS with alkyl radical in oxidation reaction, density functional theory (DFT) calculations were performed (Fig. 6). First, the initially $^1O_2$ species is reduced by the in situ-formed $Fe^{II}$ nodes, giving the active oxygen species $O_2^{•-}$, which was adsorbed by Fe–NDI to afford $Fe^{III}$–OO$^•$ species (S2). After combining a proton and an electron to form $Fe^{III}$–OOH, the total Gibbs free energy decreased by −0.39 eV (S2 to S3). The Fe-OOH species was subjected to the interaction of another iron atom in the binuclear iron node allowing its distance to be reduced to 2.6 Å to form the Fe...OO(H)–Fe transition state by overcoming an energy barrier of 0.95 eV (S3 to TS1). Subsequently, the O–O bond was cleaved by the attraction of the binuclear iron node, generating a Fe(IV)=O transition

state species with the decrease of free energy by −1.78 eV (TS1 to S4), accompanied by the oxidation of another Fe(III) to Fe(IV). It was noteworthy that the generated Fe(IV)=O bond length was 1.66 Å, which was in agreement with that of the non-heme monooxygenase[69]. Moreover, the Raman spectrum of Fe–NDI during the oxidation reaction showed a scattering peak at 820 cm⁻¹, suggesting the formation of Fe(IV)=O in Fe–NDI (Fig. S24)[70,71]. The total energy decreases further by −2.19 eV due to the interaction of the substrate radical with Fe–NDI (S4 to S5) and by −0.56 eV owing to the oxidation of the intermediate (S5 to S6). Finally, the selectively formed cyclohexanone desorbs from the catalyst, regenerating the catalyst and re-liganding the chlorine ion to the iron node (S6 to T1); this process decreased the free energy further by −0.28 eV to reach the minimum energy (Table S5, Fig. S25). In addition, dynamic tracking experiments were performed on Fe–NDI using different photon powers by adjusting the voltage or the current under standard conditions. As expected, the correlation between the product yield and photon power was consistent with a generalized quadratic model, which suggested that the photocatalytic reaction involves two photons excitation (Fig. 3h)[11]. The quadratic dependency of the product yield on the irradiation power (full power: half power; ideal 4:1) confirms the two photons excitation (Fig. 3i)[29].

This approach, developed utilizing EnT-mediated multiphoton excitation, ensures the activation of oxygen and the occurrence of oxidative transformation under light excitation in ambient air in absence of any oxidant supplementation[72–74]. This prevents the generation of toxic waste from oxidants and overcomes the issues of natural enzymes, such as instability and difficulty in recycling. This strategy is different from the traditional multiphoton excitation strategies that involve the irradiation of photosensitizers in organic dyes and coordination polymers to produce radical species through consecutive PET or consecutive PET/LMCT excitation. In our strategy, the photoactive NDI in the coordination polymer activates $O_2$ to form $^1O_2$ via EnT and the binuclear Fe(III) species activate the $C(sp^3)$–H bonds to generate radicals and mixed-valence iron species via LMCT. The Fe(II) centers in the in situ-formed mixed-valence iron species capture the $^1O_2$ to form the $Fe^{IV} = O$ intermediate, which combines the alkyl radicals to accomplish direct C–O coupling for giving the monooxygenation products in the well-modified Fe catalytic manifolds. This parallel two photons excitation strategy avoids the restrictions of the intrinsic instability and shorter lifetime of the transient state in traditional two-photon excitation systems[75,76] and is beneficial for promoting the efficacy of consecutive photoexcitation for photocatalytic transformations by simultaneously achieving the highly stable intermediate state. This heterogeneous energy transfer-mediated multiphoton excitation with kinetic synergy effect enables the selective activation of $C(sp^3)$–H via the use of a binuclear iron catalyst, enabling molecular oxygen activation and the capture of active $^1O_2$ and radical species.

## Discussion

We developed a approach for the synergistic merging of photoinduced energy transfer (EnT), ligand-to-metal charge transfer (LMCT) and hydrogen atom transfer (HAT) in one binuclear iron coordination polymer for the parallel activation of $O_2$ molecules and $C(sp^3)$–H bonds. A coordination polymer, Fe–NDI was assembled from a binuclear iron node and a functional ligand. The LMCT that affords the mixed-valence binuclear $Fe^{III}Fe^{II}$ species also produces chlorine radicals, which activate the inert alkane $C(sp^3)$–H bond via HAT. Light alkanes were directly functionalized. Subsequently, the decarboxylated functionalization of $C(sp^3)$–H were performed to further explore the $C(sp^3)$–H activation potential of Fe–NDI. Finally, the highly selective oxidation of $C(sp^3)$–H bonds is accomplished through the synergistic activation of oxygen to singlet oxygen by NDI via a two photons process. This EnT-mediated multiphoton excitation photocatalyst combines binuclear iron catalysis and multiphoton photocatalysis, as evidenced by the well-modified catalytic property and selectivity as well as the quadratic photon power dependence of Fe–NDI. We believe that our study findings highlight the potential of robust coordination polymers with uniform and precise active sites and high catalytic activity.

## Methods
### Materials and measurements

Unless otherwise noted, all the chemicals and solvents were of reagent grade quality obtained from commercial sources and used without further purification. The ligand *N,N′*-bis(5-isophthalic acid)naphthalenediimide (H₄BINDI) was synthesized according to the literature[77]. The elemental analyses of C, H, and N were performed on an Elementar UNICUBE elemental analyzer. $^1H$/$^{13}C$ NMR spectra were recorded by Vaian DLG400 with internal standard TMS at $\delta$ 0.0 ppm. ESI–MS measurements were performed on an Agilent 6224 HPLC-TOF spectrometer. Powder X-ray diffraction (PXRD) measurements were obtained on a Rigaku Smart Lab XRD instrument with a sealed Cu tube ($\lambda = 1.54178$ Å). Thermogravimetric analyses were performed on a TA Q500 instrument and recorded under $N_2$ followed by a ramp of 10 °C min$^{-1}$ up to 800 °C. Fourier transform infrared spectroscopy spectra were recorded using KBr pellets on a ThermoFisher 6700. Energy-dispersive system elemental mapping images were obtained on

a JEOL JSM-7610F Plus Field Emission Scanning Electron Microscopy. Liquid UV–vis spectra were collected on a PERSEE T9CS spectrometer. Solid UV–vis spectra were recorded on Hitachi UH5700 UV–vis–NIR spectrophotometer. Fluorescent spectra were recorded on Edinburgh FLS 1000 stable/transient fluorescence spectrometer. The EPR spectra were performed on BRUKER E500 equipped with a liquid $N_2$ system. X-ray photoelectron spectroscopy (XPS) signals were collected on a Thermo ESCALAB Xi+ spectrometer. The light source is 455 nm LED which was purchased from the Beijing China Education Au-light Co. Ltd. The gas chromatography–mass spectrometry (GC–MS) analyses were performed on Agilent Technologies 7890B GC system and Agilent 5977B MSD system. The $^{57}Fe$ Mössbauer spectroscopy was recorded on a conventional spectrometer with alternating constant acceleration of the γ-source (57Co/Rh, 0.925 GBq), which was kept at room temperature. The minimum experimental line width was 0.24 mm s$^{-1}$ (full width at half-height).

### Preparation of Fe–NDI

H₄BINDI (0.1 mmol) and FeCl₃·6H₂O (0.2 mmol) were added into high pressure reactor. The mixture added DMF (5 mL), acetic acid (0.1 mL) and H₂O (0.1 mL), then the reactor was ultrasound for ten minutes. The reactor was gradually heat up to 120 °C and within three days. After self-assembly finished, the reactor slowly cooled to room temperature. The layered yellow crystal was collected by filtration and drying. The yield of Fe–NDI was 35% based on H₄BINDI. Anal. Calcd for Fe–NDI ($C_{21}H_{19}FeN_3O_8$): C, 50.68; H, 3.85; N, 8.45%. Found: C, 50.26; H, 3.89; N, 8.31%. IR (KBr): 3083 (br, v), 1712 (vs), 1681 (m), 1586 (m), 1452 (w), 1407 (w), 1351 (s), 1284 (w), 1252 (s), 1204 (w), 1169 (w), 1122 (m), 990 (w), 767 (s), 738 (s), 683 (w), 652 (s), 542 (w), 572 (w), 415 (m) cm$^{-1}$.

### Single crystal X-ray crystallography

Intensities of Fe–NDI was collected on a Bruker SMART APEX CCD diffractometer equipped with a graphite-monochromated Mo-Kα ($\lambda = 0.71073$ Å) radiation source; the data were acquired using the SMART and SAINT programs[78,79]. The structures were solved by direct methods and refined on $F^2$ by full-matrix least-squares methods using the SHELXTL version 5.1 software[80]. In the structural refinement of Fe–NDI, all the non-hydrogen atoms were refined anisotropically. Hydrogen atoms within the ligand backbones and the coordinate DMF molecules were fixed geometrically at calculated distances and allowed to ride on the parent non-hydrogen atoms. The SQUEEZE subroutine in PLATON was used[81].

Crystal data of Fe–NDI: $C_{21}H_{19}FeN_3O_8$, $M = 497.24$, monoclinic, space group $P2(1)/c$, dark red, $a = 20.133(5)$ Å, b = 16.304(4) Å, c = 10.135(2) Å, $\alpha = \gamma = 90°$, $\beta = 103.079(5)°$, $V = 3240.7(13)$ Å$^3$, $Z = 4$, $Dc = 1.019$ g cm$^{-3}$, $\mu$(Mo-Kα) = 0.501 mm$^{-1}$, $T = 120$ K. 7610 unique reflections [$R_{int} = 0.0611$]. Final $R_1$ [with $I > 2\sigma(I)$] = 0.0700, $wR_2$ (all data) = 0.2279 for the data collected. CCDC number 2282498.

### General procedure for the C($sp^3$)–H alkylation

A 20 mL of flame-dried Schlenk quartz flask was added Fe–NDI (5.0 µmol), benzylidene malononitrile (0.1 mmol), cyclohexane (1.0 mmol) and HCl (0.05 mmol, concentrated HCl) in CH₃CN (1 mL). The resulting mixture was stirred and irradiated with a 455 nm LED under argon atmosphere at room temperature for 12 h. After the indicated time, the mixture was centrifuged at 2500$g$ for 5 min, and the supernatant was concentrated under vacuum distillation. The residues were separated on a silica gel column (EtOAc/petroleum ether) to obtain the isolated yields.

### General procedure for the decarboxylative functionalization

A 20 mL of flame-dried Schlenk quartz flask was added Fe–NDI (5.0 µmol), benzylidene malononitrile (0.1 mmol) and cyclohexanecarboxylic acid (0.50 mmol) in 1,4-dioxane (1 mL). The resulting mixture was stirred and irradiated with a 455 nm LED under argon

atmosphere at room temperature for 12 h. After the indicated time, the mixture was centrifuged at 2500$g$ for 5 min, and the supernatant was concentrated under vacuum distillation. The residues were separated on a silica gel column (EtOAc/petroleum ether) to obtain the isolated yields.

### General procedure for the C($sp^3$)−H oxidation

A 20 mL of flame-dried Schlenk quartz flask was added Fe−NDI (10.0 μmol), alkane (0.2 mmol), NH$_4$Cl (0.05 mmol) in CH$_3$CN (1.0 mL). The resulting mixture was stirred and irradiated with a 455 nm LED under air atmosphere at room temperature for 12 h. After the indicated time, the mixtures were filtered and the yields were determined by gas chromatography.

### EPR detection of reactive oxygen species

The O$_2^{\cdot-}$ and $^1$O$_2$ generated by Fe−NDI have been detected by EPR in the presence of DMPO and TEMP, respectively. For detection of O$_2^{\cdot-}$, DMPO (30 μL) in CH$_3$OH (1 mL) was mixed with CH$_3$OH (0.5 mL) suspension of Fe−NDI (1 mg). For detection of $^1$O$_2$, TEMP (30 μL) in CH$_3$CN (1 mL) was mixed with CH$_3$CN (0.5 mL) suspension of Fe−NDI (1 mg). The formed mixed solutions were drawn with capillary tubes and placed into EPR tubes. EPR measurements were carried out during the 455 nm LED light irradiation under air condition.

### Data availability

The X-ray crystallographic coordinates for the structures reported in this article have been deposited at the Cambridge Crystallographic Data Center (CCDC) under the deposition numbers CCDC 2282498 (Fe−NDI). These data can be obtained free of charge from The Cambridge Crystallographic Data Center via http://www.ccdc.cam.ac.uk/data_request/cif. All other data supporting the findings of this study are available within the article and its Supplementary Information files or from the corresponding author upon request. Source data containing figure data and atomic coordinates of the optimized structures are provided with this paper. Source data are provided with this paper.

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

## Acknowledgements

This study was supported by the National Natural Science Foundation of China (22171034 to L. Zhao and 92361201 to C. Duan), the Natural Science Foundation of Liaoning Province (2023MS116 to L. Zhao), the Natural Science Foundation of Jiangsu Province (BK20220033 to C. Duan) and the Fundamental Research Funds for the Central Universities (DUT22LAB606 to L. Zhao).

## Author contributions

Z.H.W., Y.T., and S.T.L. contributed equally to this work. L.Z. and C.Y.D. conceived the project, designed the experiments and supervised the work. Z.H.W., Y.T., and S.T.L. carried out the main experiments, collected and interpreted the data. H.Q.L. prepared the ligand. Z.H.W., S.T.L., and C.H. solved and refined the X-ray single-crystal structures. L.Z. and C.Y.D. contributed materials and analysis tools. Z.H.W., Y.T., and S.T.L. organized the data and wrote the paper. L.Z. and C.Y.D. revised the paper. All authors discussed the results and commented on the paper.

## Competing interests

The authors declare no competing interests.
