## [Peer Review File · Nature Communications]

Energy transfer-mediated multiphoton synergistic excitation for selective C(sp³)-H functionalization with coordination polymerReviewers' Comments:

Reviewer #1:

Remarks to the Author:

The work reports the activation and selective oxidation of inert C-H bonds by employing a photocatalyst involving a NDI-based ligand and diiron complex. The substrate scope is large and moderate to high yields of the products are obtained. In terms of synthetic application the work is good and is suitable for Nature Communication. However, the proposed mechanism is mainly a hypothesis presently and not supported by sufficient experimental evidences. The involvement of iron(IV)oxo moiety in the reaction, in particular, is extremely dubious. Many of the reactions proposed in Figure 1 have no precedents in the iron(IV)oxo chemistry. For example Fe(IV)=O cores generally perform HAT from R-H to form R(radical) and Fe(III)-OH cores. No evidence of the formation of Fe(III)-OR is known. Also the EPR data makes no sense. The Proposed diiron(III) species should be EPR silent, because of coupling of the two iron(III) centers, and only upon photoradiation signal should arise because of the mixed valent EPR active Fe(III) Fe(II) species. Also why no chlorination products are formed? what's the origin for the ketone selectivity.

So although the study is highly interesting the authors need to do a better job in performing the mechanistic investigation. At least some of the proposed intermediates should be properly and unambiguously characterized. Moessbauer spectroscopy may be employed to characterize the Fe(III) Fe(III) and Fe(III) Fe(II) states. All the EPR studies should be complemented with spin quantification. In the mass spectral studies for the analysis of the products, the whole spectrum in a wider range should be shown.

Reviewer #2:

Remarks to the Author:

This manuscript by Duan and co-workers described a new synthetic platform for selective C(sp³)-H functionalization under mild conditions via photoinduced EnT, LMCT and HAT process. In this paper, Fe(III) and chlorine ions could provide the mixed-valence Fe(III)Fe(II) species and chlorine radicals (Cl•) via ligand-to-metal charge transfer (LMCT), than the Cl• radicals abstract a hydrogen atom from the inert C(sp³)-H bond of alkanes via hydrogen atom transfer (HAT). In addition, the authors founded that NDI converts oxygen to 1O₂ via energy transfer (EnT), which then coordinates to Fe(II), forming an Fe(IV)=O intermediate for the selective oxidation of C(sp³)-H bonds. This approach displays broad functional group tolerance and convenient reaction conditions for C(sp³)-H bond activation and selective oxidation. Mechanistic studies reveal that the C(sp³)-H bond activation via radical formation and the C-H bond cleavage is the rate-determining step in the C-H alkylation process.

Advantage of this manuscript: As these authors stated, this manuscript represented the first example of a coordination polymer that combines photoinduced EnT and LMCT and provides a parallel excitation strategy with kinetic synergy effect of C-H bond activation and oxidation. These studies may provide a chance to recycle such catalysis in industry, of course more studies need to

be made?

Disadvantage of this manuscript: Many studies about C-H bond activation and oxidation (methane, toluene and inert C-H bonds) via photoredox catalysis have been reported in organic syntheses. Of course, most of these catalysis were structure simple, such simple Fe and Ce catalysis. This reviewer found highly related reports should be cited or highlighted in the future here or there, such as Y.-H. Wang, Q. Yang, P. J. Walsh, E. J. Schelter, *Organic Chemistry Frontiers* 2022, 9, 2612-2620; Q. Yang, Y.-H. Wang, Y. Qiao, M. Gau, P. J. Carroll, P. J. Walsh, E. J. Schelter, *Science* 2021, 372, 847-852; J. Wu, J. Chen, L. Wang, H. Zhu, R. Liu, G. Song, C. Feng, Y. Li, *Green Chemistry* 2023, 25, 940-945. Z.-X. He, B. Yin, X.-H. Li, X.-L. Zhou, H.-N. Song, J.-B. Xu, F. Gao, *Journal of Organic Chemistry* 2023, 88, 4765-4769. Based on previous studies, the authors should state what's the advantages or disadvantages of the method developed here, so that the potential readers could quickly know the backgrounds.

In summary, this work might be of the general interest of Nat. Commun. readers. This reviewer suggested its publication of this manuscript in Nat. Commun. but after addressing the following issues.

1. The structure of compounds in this paper need to be standardized, such as Table 1, 5i and 5j.
2. Please maintain consistency in the expression of yield in Tables 1 and 2.
3. For the substrate scope of alkylation in Table 1, have you tried isobutene, because isobutane can be used as starting material to prepare tertiary alkane compounds.
4. Please check the references in the manuscript, some references have formatting errors, such as ref. 23 and ref. 54.
5. For Supplementary information: The NMR spectra of the compound are too blurry. The author should provide sufficiently clear NMR spectra in the revised Supplementary information.
6. More related reports should be added in the manuscript other than the refs mentioned above.
7. Could the authors isolate the chloro radical added products using other chloro acceptors, such as Ts protected bis allyl amines or hepta-1,6-diene .
8. What happened for simple toluene oxidation. Could the authors make benzaldehydes by using toluene. So far, aldehydes is difficult to make by using photoredox chemistry. The authors should give these results in the manuscripts

Reviewer #3:

Remarks to the Author:

Ubiquitous to the strongest oxidation catalysts in nature and benchtop chemistry is the generation of ROS species and/or high-energy metal-oxo/peroxo intermediates. Analysis of these systems has been ongoing for at least 150 years. The ability of heme- and non-heme iron enzymes like cytochrome p450s and sMMO to oxidize strong C-H bonds through HAT mechanisms has inspired the development of synthetic analogues whose study has perplexed researchers for over a century. In this submission, the authors have presented an intriguing novel system reportedly utilizing two synergistic photoexcitation phenomena to achieve C—H bond activation by a binuclear iron

catalyst embedded in a crystalline coordination polymer. The authors propose the complementary charge transfers from two distinct manifolds yield a potent oxidation catalyst reminiscent of iron-based monooxygenases. The proposed mechanism is intriguing with coupled photoexcitation from two manifolds yielding impressive radical coupling reactivity. However, dramatic inconsistencies between figures, text, and supporting information provided raise several questions regarding the mechanistic details. Further scrutiny leads this reviewer to question the role of the title species in the photocatalysis, as nearly identical reactivity has been shown by a simple homogeneous system cited by the authors:

Dai, Z.-Y.; Zhang, S.-Q.; Hong, X.; Wang, P.-S; Gong, L.-Z. A practical FeCl₃/HCl photocatalyst for versatile aliphatic C-H functionalization. *Chem Catal.* 2022, 2, 1211-1222.

The absence of the obvious control experiments towards unambiguous exclusion of the cited potential homogeneous active photocatalyst potentially generated by partial decomposition of the heterogeneous system (as shown by the authors) and identical reactivity evaluated under nearly identical conditions is a major oversight by the authors and I cannot therefore recommend this work for publication in *Nature Communications*. Detailed concerns are listed below:

1) Of critical importance to much of this manuscript is the concentration of HCl used in the various experiments. It is unclear whether an aqueous or other HCl source was used from the Materials and Methods section. If the source is aqueous, additional complexity is added due to the known hydrolysis of MeCN in the presence of concentrated aqueous HCl. Additionally, it is unclear how the pH was measured in the HCl/MeCN solutions. Clear discoloration of solutions below the reported pH=2 is evident, along with a reported 10-15% loss of catalyst. This coupled with the known photocatalytic behavior of FeCl₃/HCl/MeCN solutions which generate identical products under 390nm LED irradiation under aerobic/anaerobic conditions, like the data presented. On page 8, line 209, the authors claim no FeCl₃ is present in the system, which is clearly refuted by figures 2a and S13, where diagnostic LMCT bands for FeCl₄⁻ are observed in the UV-Vis, and immediately preceding this statement on page 8, line 205, "Fe-NDI was removed from the reaction system by filtration after 4, 6, and 8 hours of the photocatalytic reaction and the filtrate continued to react for up to 14 hours under standard conditions."

Thus, only the chloride-free decarboxylative functionalization reaction appears to be genuinely catalyzed by Fe-NDI, however, no LMCT bands were observed, precluding a similar mechanism to that proposed by the authors and similar loss of the diagnostic Fe-NDI absorption above 375nm. It is possible that the fluorescence bands observed between 490-520 nm are consistent with free H4BINDI ligand excited at 455nm. As solid-state emission profiles of free H4BINDI ligand show a strong band between 416-530nm assigned to an intraligand $\pi \rightarrow \pi^*$ transition when excited at 390nm. *Inorg. Chem.* 2023, 62, 6661-6673

Therefore, even for the decarboxylative functionalization reactions, the active catalyst is potentially a homogeneous Fe(III) species.

Because the authors have not unambiguously excluded a homogeneous photocatalyst, have shown decomposition of the heterogeneous photocatalyst under reaction conditions, and the likely homogeneous catalyst has already been reported for two of the three reactions presented, the intriguing coupled EnT/LMCT mechanism is not sufficiently supported in this system.

2) The net reaction detailed in Fig. 1a and the object of the computational inquiry (Fig. 4, Table S5,

and Fig. S17) is:

and that this reaction occurs at an iron catalyst, is at heart an example of Fenton chemistry. I encourage the authors to consult the following reference:

Barton, D. H. R.; Dollar, D. *Acc. Chem. Res.* 1992, 25, 504-512.

Therein, the authors will find similar observations regarding the absence of evidence for OH^* , as is commonly invoked in Fenton mechanisms, as well as an alternative mechanism for the same overall oxidation of cyclohexane. As Fenton chemistry generally involves H_2O_2 as a reagent, the authors should exclude its involvement with a H_2O_2 -specific quenching agent, e.g. catalase or $\text{CuSO}_4/2,9$ -dimethyl-1,10-phenanthroline, as DMPO and Benzoquinone do not distinguish between $\text{O}_2^{\cdot-}$ and H_2O_2 .

3) There are a number of inconsistencies between what is shown in Fig. 1, Fig. 4, Table S5, Fig. S17, and the text relating to the details of the mechanism of the substrate (probably better described as RH_2 for consistency) oxidation reaction.

Panel (a) in Fig. 1 shows the oxidation of the FeII site of the catalyst by 1O_2 to form an FeIII-OH intermediate, neglecting an H^* source. This is followed by the evolution of H_2O from FeIII-OH to yield the key FeIV=O species. Heterolytic cleavage of the FeIII-O—OH bond after the addition of a proton, would yield an FeV=O species, while homolytic cleavage of the FeIII-O—OH bond would yield the FeIV=O species presented. Therefore, in the homolytic case presented, another source of net H^* is required from the scheme to generate both the FeIII-OH and the H_2O . However, the figure and text specify Cl^* generated from a separate photolytic pathway is performing the hydrogen abstraction from the RH_2 substrate to yield an RH^* radical. The subsequent addition of the RH^* radical to the FeIV=O yields the FeIII-O-RH intermediate. Finally, the loss of a proton is shown to generate an FeIII-O-R * radical species which disobeys charge balance. Finally exchange with a Cl^- anion yields FeIII-Cl and the R=O product, regenerating the catalyst. If Cl^* is performing HAT to generate RH^* , Cl^- , and H^+ , there is no source of H^* in Fig. 1a which is a critical oversight for at least two of the steps shown.

This critique is corroborated by the authors' own DFT calculations as shown in Fig. 4. Here the authors explicitly show the addition of an exogenous electron between intermediates S1 and S2 and the addition of an exogenous proton between steps S2 and S3. However, in this figure, the authors again neglect a net H^* source somewhere between S3 and S4 to generate H_2O as well as not noting the addition of the exogenous chloride anion in steps S6 to T1. Shown in the reaction path, there is an addition of a net H^* and a loss of a net H^* but this still leaves the evolution of H_2O unbalanced. Ostensibly, the initial electron source could originate from the photoexcited NDI^* shown at the top left of figure 1, thus allowing the acidic solution (NH_4Cl in MeCN) to provide the proton and leaving a positive ligand to be reduced to close the catalytic cycle, yet this still doesn't account for the evolution of water.

4) There are some apparent errors in the spin states of the calculations as shown in Fig. 4. Addition of singlet O_2 and a single electron to S1 (presumably "singlet 1" despite being composed of a d5 FeIII and a d6 FeII) to form S2 (presumably "singlet 2") is impossible as addition of a single electron to a singlet state must result in even multiplicity, with a doublet the expected spin state. If Table S5 is to be believed, which differs from Fig. 4, then there is an error between S2 and S3 where a net

hydrogen atom is added to the system which would necessarily result in an even multiplicity spin state. If S and T do not refer to singlet and triplet spin states respectively, this should be explicitly stated somewhere.

Furthermore, the fact that the FeIII centers of pristine Fe-NDI are high-spin, as evidenced by EPR, complicates spin-state assignment. The FeIII sites in the crystal structure are bridged by two carboxylate ligands, which are well known to behave as superexchange pathways, resulting in some antiferromagnetic coupling of the iron centers, which can be expected to at least partially quench the EPR signal. It would be helpful to quantify the (total spin/total Fe) of the samples in the EPR chamber. It could reveal either enhanced antiferromagnetic coupling upon the addition of CHCA, HCl and NH₄Cl, which would be consistent with the resistivity measurements, or the generation of low-spin FeIII.

5) The scheme reported in the SI (Table S5) is inconsistent with Fig. 4, showing instead that the S1 to S2 reaction does not involve the addition of an electron (also, in Table S5 all “ST1” should be “TS1” to be consistent with Fig. 4, Fig. S17, and the text). The authors did not provide coordinates for any of the 8 optimized structures, nor a table of relevant structural parameters, which precludes identification of reasonable parameters to assist in correcting this error (e.g., is the O-O bond length in S2 reasonable for a superoxo or a peroxide?). Moreover, by eye the Fe—O₂ bond length appears to be exceptionally long. Assuming the other Fe—O bond lengths are similar to those reported in the crystal structure parameter table, the Fe—O₂ bond must be greater than the ~1.9-2.1 Å of reported FeIII-superoxos. As shown in Fig. 4, the text states that a proton is added to S2 to generate S3, which again is inconsistent with table S5 which states hydrogen and electron addition occurs at this step.

6) Assuming Table S5 is the most correct, it is very odd that a net hydrogen atom is added to the complex between the transition state and S4. This begs the question of how the authors define a transition state. Distortion in either direction along the imaginary mode of the transition state should lead to reactants or products, but never have I seen an example of the addition of exogenous species to a transition state to generate products. The authors don't specify how this TS was acquired, e.g. nudged elastic band is a common method on VASP, though how one would add a net hydrogen atom from space is unclear.

7) While it is clear there is some change occurring to the Fe sites of Fe-NDI in the presence of Cl- upon exposure to HCl or NH₄Cl based on the UV-Vis data (Fig. 2a), evidence of coordination of Cl- to the Fe sites would be better presented by highlighting the emergence of the 2 strong absorption bands (Cl⁻FeIII LMCT) at ~310 and ~360nm, and the lack thereof in the CHCA spectrum (S15). Adding an analogous supporting UV-Vis spectra for the NH₄Cl case would make this claim unambiguous if those same two LMCT bands are observed. The evolution of those two bands in the presence of HCl are very similar to those observed in solutions of FeCl₃ at high HCl concentrations: *Microchimica Acta*, 2020, 187, 488 and high LiCl concentrations: *Chem. Geol.* 2006, 231, 326-349 moreover, those two bands are diagnostic of FeCl₄⁻: *J. Am. Chem. Soc.* 1953, 75, 1435-1443

and have been assigned to FeCl₄⁻ LMCT bands; which can persist upon ligation, coincident with notable red-shift of the ligand $\pi \rightarrow \pi^*$ band:

Open J. Inorg. Chem. 2013, 3, 7-13

8) The authors need to state somewhere what excitation wavelength was used for the fluorescence experiments (Fig. 2a), if the 455nm LED was used, it would also be beneficial to provide a spectrum showing the bandwidth of this source.

9) In the Table 1 caption for conditions d, "CHCA" should be "4", and in the description, line 265, page 11, "5a" should be bold.

10) Unless the EPR and XPS spectra were collected during in-situ irradiation (if so, this should be explicitly stated, and dark control conditions should be presented), the EPR and XPS findings to support partial reduction of the system to generate Fe^I is unconvincing. The appearance of the Fe-NDI + HCl/CHCA peaks assigned as Fe^I 2p_{1/2} coincides with a drastic ~+3 eV shift of the assigned Fe^{III} 2p_{1/2} peaks. While there is clearly an emergence of additional features in the XPS spectra, it is far more likely attributable to multiple Fe^{III} environments due to coordination of Cl⁻ or cyclohexanecarboxylate. Moreover, the absence of an increased absorbance peak for the CHCA soaked sample near 455nm (Fig. S15) makes it unlikely that LMCT from coordinated CHCA is resulting in the reduction of Fe^{III}. The diminished EPR signal could just as easily be assigned to a conversion from high-spin to low-spin Fe^{III} due to a change in coordination environment or as a consequence of enhanced antiferromagnetic coupling as mentioned above. In the absence of the higher-field of the EPR data, the reduction of Fe^{III} cannot be concluded from this data, only a lower concentration of high-spin Fe^{III}.

11) On page 7, line 175, the authors suggest the decreased resistance is a consequence of Cl⁻ coordination; however, the lowest resistance was achieved with the chloride-free acid CHCA. Thus, it appears more likely that ligation change as a consequence of acid exposure is the most logical origin of decreased resistance.

12) On page 8, lines 204-210 are very confusing; the authors state that after removing the heterogeneous photocatalyst, "the filtrate continued to react for up to 14 hours under standard reaction conditions", but the catalytic yield is unchanged, and Fig. S13 shows that the yield at 4, 8, and 16 hours as unchanged. This sentence should be revised. Either i) the reaction stops after removal of the photocatalyst, ii) a change occurs as a result of imperfect filtration, or iii) the active photocatalyst is homogeneous.

13) On line 288, the authors suggest both ¹O₂ and O₂^{-•} were observed from radical trapping agents, but DMPO is promiscuous and O₂^{-•} and O₂²⁻ (H₂O₂) are indistinguishable. Catalase is commonly used as an H₂O₂-selective quencher in similar cases across a range of solvents including MeCN, and can tolerate mildly acidic conditions.

14) The acronym CHCA needs to be defined earlier in the manuscript text or in the caption of Fig. 2 and Table 1. CHCA is not defined until page 11, while it is referenced in all but one panel of Fig. 2

and Table 1 currently on pages 5 and 9, respectively. The first time a panel that includes CHCA is discussed in the text is on line 100 (page 4), and without that definition, it is difficult to contextualize the findings.

15) Caption of Fig. S7 should change “The first weightlessness could belong to the loss of free and coordination solvent. The second weightlessness was attributed to the decomposition of Fe-NDI.” to “The first weight loss event could be due to the loss of free and coordinated solvent. The second weight loss event was attributed to the decomposition of Fe-NDI.”

16) Figure S11 x-axis label should be “amount” instead of “mount”.

17) Figures S15 and S16 captions need to be more detailed w.r.t. concentrations of species other than CHCA, and Fig. S16 caption needs to include excitation wavelength.

Reviewer 1:

Comments: The work reports the activation and selective oxidation of inert C–H bonds by employing a photocatalyst involving an **NDI**-based ligand and diiron complex. The substrate scope is large and moderate to high yields of the products are obtained. In terms of synthetic application, the work is good and is suitable for Nature Communication. However, the proposed mechanism is mainly a hypothesis presently and not supported by sufficient experimental evidences. The involvement of iron(IV)oxo moiety in the reaction, in particular, is extremely dubious. Many of the reactions proposed in Figure 1 have no precedents in the iron(IV)oxo chemistry. For example, Fe(IV)=O cores generally perform HAT from R–H to form R(radical) and Fe(III)–OH cores. No evidence of the formation of Fe(III)–OR is known. Also the EPR data makes no sense. The Proposed diiron(III) species should be EPR silent, because of coupling of the two iron(III) centers, and only upon photoradiation signal should arise because of the mixed valent EPR active Fe(III) Fe(II) species. Also why no chlorination products are formed? what's the origin for the ketone selectivity.

So although the study is highly interesting the authors need to do a better job in performing the mechanistic investigation. At least some of the proposed intermediates should be properly and unambiguously characterized. Moessbauer spectroscopy may be employed to characterize the Fe(III) Fe(III) and Fe(III) Fe(II) states. All the EPR studies should be complemented with spin quantification. In the mass spectral studies for the analysis of the products, the whole spectrum in a wider range should be shown.

Responses: Many thanks to the reviewer for the positive and kind comments. In this study, the efficient activation and selective oxidation of C(*sp*³)–H bonds were successfully implemented via energy transfer-mediated multiphoton synergistic excitation by Fe–**NDI** integrating the Fe(III)–Cl chromophore LMCT process and **NDI** activation oxygen through the energy transfer process. The LMCT process of the Fe(III)–Cl chromophore to form the chlorine radical for C(*sp*³)–H bonds activation via HAT was verified using spectroscopy, mass spectrometry and efficiently photocatalytic C(*sp*³)–H bond alkylation model reaction. Moreover, EPR tests were considered to characterize the change of valence state of Fe nodes in Fe–**NDI** after irradiation with a 455 nm LED to further validate the occurrence of the LMCT process. But, as the reviewer said, the EPR signals of the binuclear Fe(III) species would be silenced due to the antiferromagnetic coupling effect, while the weak signals in the current EPR spectra might be caused by the test environment and the slight changes in EPR signals were not enough to support the conclusion that LMCT occurs. Mössbauer spectroscopy is an ideal and accurate method to selectively probe the geometry and purity of Fe environments in a sample. According to the reviewer's suggestion, the zero-field ⁵⁷Fe Mössbauer spectrum of Fe–**NDI** tests was executed at 80 K. The results showed a symmetric doublet with isomer shift $\delta = 0.51 \text{ mm s}^{-1}$ and quadrupole splitting $|\Delta E_Q| = 0.81 \text{ mm s}^{-1}$, which was attributed to the high-spin Fe(III) species in Fe–**NDI** (Figure 2f). Irradiation of the Fe–**NDI**

acetonitrile suspension containing HCl with a 455 nm LED gave a mixed peak pattern, on the zero-field ^{57}Fe Mössbauer spectrum. Through data fitting, the red peak pattern shows isomer shift $\delta = 0.52 \text{ mm s}^{-1}$ and quadrupole splitting $|\Delta\text{EQ}| = 0.83 \text{ mm s}^{-1}$, which was attributed to a high-spin Fe(III) iron species. The blue peak pattern shows isomer shift $\delta = 1.32 \text{ mm s}^{-1}$ and quadrupole splitting $|\Delta\text{EQ}| = 3.10 \text{ mm s}^{-1}$, which was attributed to a high-spin Fe(II) iron species. A 27% conversion from the high-spin Fe(III) species to high-spin Fe(II) species in Fe-**NDI** under the irradiation of a 455 nm LED further verifies the occurrence of the LMCT process. The formed highly electrophilic Cl radical can grab the H atom from the C-H bond of alkanes to generate alkyl radicals for participating in the subsequent functionalization process, accompanied by Fe(II) formation.

Synergistic with the LMCT process was the activation of molecular oxygen to singlet oxygen ($^1\text{O}_2$) achieved by Fe-**NDI** through energy transfer. The formation of $^1\text{O}_2$ was identified by TMB oxidation experiments, EPR tests and α -terpinene oxidation experiments. The Fe(II) combined singlet oxygen with the assistance of protons and electrons to form the highly valent Fe(IV)=O unit, a highly reactive intermediate observed in Fe-**NDI** after the catalytic oxidation reaction by the Raman spectroscopy (Figure S22). It is generally acknowledged that Fe(IV)=O cores directly extract the H atom from R-H via HAT to form $^{\bullet}\text{RH}$ and Fe(III)-OH cores, which subsequently selectively produce the corresponding alcohols. In the present study, before the Fe(IV)=O cores formed, $^{\bullet}\text{RH}$ had been created via HAT by the Cl radical produced after the LMCT process of the Fe(III)-Cl unit, which could be captured directly by Fe(IV)=O to selectively produce the corresponding keto products via Fe-O-RH intermediate subsequently. Even the small amount of alcohol products produced via the general activation process during oxidation would continue to be further oxidized to keto products. In order to verify this proposed mechanism, DFT theoretical calculations were carried out. The results showed that the existence of Fe(IV)=O and Fe(III)-O-RH paths was thermodynamically spontaneous, suggesting the unique advantage of Fe-**NDI** two photons excitation synergizing the LMCT process with the EnT process for highly selective catalytic C-H oxidation to ketone products.

Due to the high efficiency of this photocatalytic system for C-H bond activation and transformation, the concentration of by-products is too low to be separated to calculate the isolated yield. So we are very sorry that we didn't verify whether there were chlorination products. According to the reviewer's suggestion, we repeat the alkylation and oxidation experiments and use GC-MS to detect whether chlorination products are formed. The results showed only traces of the chlorinated products were detected. In fact, the chlorination products were also rarely mentioned in numerous researches where the metal-Cl chromophores LMCT process released Cl radicals to complete C-H bond activation (*Science* 2021, **372**, 847-852; *Org. Chem. Front.* 2022, **9**, 2612-2620; *J. Am. Chem. Soc.* 2021, **143**, 2729-2735), possibly due to the fact that alkyl radicals were more likely to combine with radical trapping agents to form the corresponding

reaction products. Moreover, the mass spectra of adducts from chlorine radical and alkyl radical capture experiments were retested, and the whole spectra in a wider range were provided in revised supporting information. The Mössbauer spectrum of Fe-**NDI** before and after the oxidation reaction and related descriptions were also added to the revised manuscript.

Reviewer 2:

Comments: This manuscript by Duan and co-workers described a new synthetic platform for selective C(sp^3)-H functionalization under mild conditions via photoinduced EnT, LMCT and HAT process. In this paper, Fe^{III} and chlorine ions could provide the mixed-valence Fe^{III}Fe^{II} species and chlorine radicals (Cl[•]) via ligand-to-metal charge transfer (LMCT), then the Cl[•] radicals abstract a hydrogen atom from the inert C(sp^3)-H bond of alkanes via hydrogen atom transfer (HAT). In addition, the authors found that NDI converts oxygen to ¹O₂ via energy transfer (EnT), which then coordinates to Fe^{II}, forming an Fe^{IV}=O intermediate for the selective oxidation of C(sp^3)-H bonds. This approach displays broad functional group tolerance and convenient reaction conditions for C(sp^3)-H bond activation and selective oxidation. Mechanistic studies reveal that the C(sp^3)-H bond activation via radical formation and the C-H bond cleavage is the rate-determining step in the C-H alkylation process.

Advantage of this manuscript: As these authors stated, this manuscript represented the first example of a coordination polymer that combines photoinduced EnT and LMCT and provides a parallel excitation strategy with kinetic synergy effect of C-H bond activation and oxidation. These studies may provide a chance to recycle such catalysis in industry, of course more studies need to be made?

Disadvantage of this manuscript: Many studies about C-H bond activation and oxidation (methane, toluene and inert C-H bonds) via photoredox catalysis have been reported in organic syntheses. Of course, most of these catalysis were structure simple, such as simple Fe and Ce catalysis. This reviewer found highly related reports should be cited or highlighted in the future here or there, such as Y.-H. Wang, Q. Yang, P. J. Walsh, E. J. Schelter, *Organic Chemistry Frontiers* 2022, 9, 2612–2620; Q. Yang, Y.-H. Wang, Y. Qiao, M. Gau, P. J. Carroll, P. J. Walsh, E. J. Schelter, *Science* 2021, 372, 847–852; J. Wu, J. Chen, L. Wang, H. Zhu, R. Liu, G. Song, C. Feng, Y. Li, *Green Chemistry* 2023, 25, 940–945. Z.-X. He, B. Yin, X.-H. Li, X.-L. Zhou, H.-N. Song, J.-B. Xu, F. Gao, *Journal of Organic Chemistry* 2023, 88, 4765–4769. Based on previous studies, the authors should state what's the advantages or disadvantages of the method developed here, so that the potential readers could quickly know the backgrounds.

In summary, this work might be of the general interest of Nat. Commun. readers. This reviewer suggested its publication of this manuscript in Nat. Commun. but after addressing the following issues.

Response: Many thanks to the reviewer for the positive and kind comments. We will try our best to improve the quality of our publications. According to the reviewer's suggestion, we have added citations to the relevant literature and summarised the advantages of this article to help readers understand the context of this research.

Reviewer's Comments (1): The structure of compounds in this paper needs to be standardized, such as Table 1, 5i and 5j.

Responses: Thanks very much to the reviewer for the careful suggestions and we are sorry for this unprecise depiction. We have corrected the standard structure of these two compounds and carefully checked the full manuscript and supporting information to ensure the accuracy of all structures of compounds.

Reviewer's Comments (2): Please maintain consistency in the expression of yield in Tables 1 and 2.

Responses: Thanks very much to the reviewer for the careful suggestions. The word "Yield" in Table 2 has been removed to maintain consistency in the expression of yield in Tables 1 and 2.

Reviewer's Comments (3): For the substrate scope of alkylation in Table 1, have you tried isobutene, because isobutane can be used as starting material to prepare tertiary alkane compounds.

Responses: Many thanks to the reviewer for the careful suggestions. According to the reviewer's suggestion, we have attempted to supplement the photocatalytic C(*sp*³)-H bond alkylation experiment using isobutane as a substrate. The irradiation of a CH₃CN (7.0 mL) solution containing Fe-NDI (10.0 μmol), benzylidenemalononitrile (0.1 mmol), HCl (0.1 mmol) and balloon with isobutane with a 455 nm LED at room temperature in argon for 48 hours produced the target product with a yield of approximately 50% and the ratio of the primary carbon alkylation product and the tertiary carbon alkylation product is 1:3.3. The result demonstrate that this photocatalytic C(*sp*³)-H bond activation system is also suitable for the conversion of isobutane.

Reviewer's Comments (4): Please check the references in the manuscript, some references have formatting errors, such as ref. 23 and ref. 54.

Responses: Thanks very much for the reviewer's suggestions and we are very sorry for the formatting problems in the references. We have corrected the formatting of these two references and carefully checked the full manuscript and supporting information to ensure the accuracy of all reference formatting.

Reviewer's Comments (5): For Supplementary information: The NMR spectra of the compound are too blurry. The author should provide sufficiently clear NMR spectra in the revised Supplementary information.

Responses: Thanks very much to the reviewer for suggestions. The unclear NMR spectra of all the compounds have been updated for identification in the revised supplementary information.

Reviewer's Comments (6): More related reports should be added in the manuscript other than the refs mentioned above.

Responses: Many thanks to the reviewer for the careful suggestions. According to the reviewer's suggestion, we have added more related references to discuss the advantages and innovations of this study in the revised manuscript.

Reviewer's Comments (7): Could the authors isolate the chloro radical added products using other chloro acceptors, such as Ts protected bis allyl amines or hepta-1,6-diene.

Responses: Many thanks to the reviewer for the thoughtful suggestions. The incorporation of radical traps in the reaction system to capture chloro radicals is an important method of validating the reaction pathway. In this article, the chloro radical added products were detected by adding benzylidenemalononitrile and styrene, suggesting that chloro radicals were involved in the reaction process. Furthermore, at the suggestion of the reviewer, Ts protected bis allyl amine as another chloro acceptor was added into the alkylation model reaction catalyzed by Fe–NDI. The products of chloro radical trapping experiment were isolated and analyzed by liquid chromatography-mass spectrometry. The result displayed two isotopic peaks at $m/z = 857.6121$ and 1143.1476 , which were reasonably assigned to $[\text{C}_{13}\text{H}_{18}\text{ClNO}_2\text{SNa}]^+$ and $[\text{C}_{13}\text{H}_{17}\text{Cl}_2\text{NO}_2\text{SNa}]^+$ species via a comparison with the simulation result based on natural isotopic abundances, respectively (Figure S14), further indicating the formation of chlorine radicals. We have added some words to describe it in the revised manuscript and the data have been added to the revised supporting information.

Reviewer's Comments (8): What happened for simple toluene oxidation. Could the authors make benzaldehydes by using toluene. So far, aldehydes are difficult to make by using photoredox chemistry. The authors should give these results in the manuscripts.

Responses: Many thanks to the reviewer for the useful suggestions. Toluene is also an important reaction substrate in the C–H bond oxidation reaction. As the reviewer said, benzaldehydes are difficult to make from toluene by using photoredox chemistry, especially by the light-mediated hydrogen atom transfer strategy, because the bond dissociation energies of C–H bond of aldehyde group (BDE ≈ 88 kcal/mol) on benzaldehyde and methyl group (BDE ≈ 85 kcal/mol) on toluene are much lower than that of the Cl–H bond (103 kcal/mol), benzaldehyde is easily further oxidized to benzoic acid. As the suggestion of the reviewers, we supplemented the toluene oxidation experiment under the same conditions. As expected, the yields of benzaldehyde and benzoic acid were 10% and 63%, respectively. We have added some words to describe this result in the revised manuscript.

Reviewer: 3

Comments: Ubiquitous to the strongest oxidation catalysts in nature and benchtop chemistry is the generation of ROS species and/or high-energy metal-oxo/peroxo intermediates. Analysis of these systems has been ongoing for at least 150 years. The ability of heme- and non-heme iron enzymes like cytochrome p450s and sMMO to oxidize strong C–H bonds through HAT mechanisms has inspired the development of synthetic analogues whose study has perplexed researchers for over a century. In this submission, the authors have presented an intriguing novel system reportedly utilizing two synergistic photoexcitation phenomena to achieve C–H bond activation by a binuclear iron catalyst embedded in a crystalline coordination polymer. The authors propose the complementary charge transfers from two distinct manifolds yield a potent oxidation catalyst reminiscent of iron–based monooxygenases. The proposed mechanism is intriguing with coupled photoexcitation from two manifolds yielding impressive radical coupling reactivity. However, dramatic inconsistencies between figures, text, and supporting information provided raise several questions regarding the mechanistic details. Further scrutiny leads this reviewer to question the role of the title species in the photocatalysis, as nearly identical reactivity has been shown by a simple homogeneous system cited by the authors:

Dai, Z.-Y.; Zhang, S.-Q.; Hong, X.; Wang, P.-S.; Gong, L.-Z. A practical FeCl₃/HCl photocatalyst for versatile aliphatic C–H functionalization. *Chem Catal.* 2022, 2, 1211–1222.

The absence of the obvious control experiments towards unambiguous exclusion of the cited potential homogeneous active photocatalyst potentially generated by partial decomposition of the heterogeneous system (as shown by the authors) and identical reactivity evaluated under nearly identical conditions is a major oversight by the authors and I cannot therefore recommend this work for publication in *Nature Communications*. Detailed concerns are listed below:

Reviewer's Comments (1): Of critical importance to much of this manuscript is the concentration of HCl used in the various experiments. It is unclear whether an aqueous or other HCl source was used from the Materials and Methods section. If the source is aqueous, additional complexity is added due to the known hydrolysis of MeCN in the presence of concentrated aqueous HCl. Additionally, it is unclear how the pH was measured in the HCl/MeCN solutions. Clear discoloration of solutions below the reported pH = 2 is evident, along with a reported 10-15% loss of catalyst. This coupled with the known photocatalytic behavior of FeCl₃/HCl/MeCN solutions which generate identical products under 390 nm LED irradiation under aerobic/anaerobic conditions, like the data presented. On page 8, line 209, the authors claim no FeCl₃ is present in the system, which is clearly refuted by figures 2a and S13, where diagnostic LMCT bands for FeCl₄⁻ are observed in the UV-Vis, and immediately preceding this statement on page 8, line 205, “Fe–NDI was removed from the reaction system by filtration after 4, 6, and 8 hours of the photocatalytic reaction and the filtrate continued to react for up to 14 hours under standard

conditions.”

Thus, only the chloride-free decarboxylative functionalization reaction appears to be genuinely catalyzed by Fe–**NDI**, however, no LMCT bands were observed, precluding a similar mechanism to that proposed by the authors and similar loss of the diagnostic Fe–**NDI** absorption above 375 nm. It is possible that the fluorescence bands observed between 490–520 nm are consistent with free H₄**BINDI** ligand excited at 455 nm. As solid-state emission profiles of free H₄**BINDI** ligand show a strong band between 416–530 nm assigned to an intraligand π - π^* transition when excited at 390 nm.

Inorg. Chem. 2023, 62, 6661–6673

Therefore, even for the decarboxylative functionalization reactions, the active catalyst is potentially a homogeneous Fe^{III} species.

Because the authors have not unambiguously excluded a homogeneous photocatalyst, have shown decomposition of the heterogeneous photocatalyst under reaction conditions, and the likely homogeneous catalyst has already been reported for two of the three reactions presented, the intriguing coupled EnT/LMCT mechanism is not sufficiently supported in this system.

Responses: Thanks very much to the reviewer for careful suggestions and we are sorry for this negligence of not describing clearly the HCl source and how the pH was measured in the HCl/MeCN solutions. In this manuscript, all catalytic experiments were performed in acetonitrile solutions containing trace amounts of concentrated hydrochloric acid (12 M), and no additional water was added, so as to ensure sufficient chloride ion concentration to form Fe–Cl chromophore and avoid the potential hydrolysis of acetonitrile as mentioned by the reviewer. In addition, the mixture of concentrated hydrochloric acid (4 μ L) and acetonitrile (1 mL) from the catalytic condition was examined by GC-MS and no acetonitrile hydrolysis products were detected. For how to determine the pH in the HCl/MeCN solutions: we first prepared aqueous solutions with pH values of 1 to 9 using concentrated hydrochloric acid and deionized water, and then mix with acetonitrile to prepare a mixed solution to test the stability of Fe–**NDI**. We have updated the description accordingly in the revised manuscript and supporting information.

In this study, inspired by the efficient activation of the C–H bond by FeCl₃ to form chlorine radical through the LMCT process, chloride ions were introduced to combine with Fe–**NDI** for constructing Fe–Cl chromophore as a catalytic site for C–H bond activation and transformation. In fact, as the reviewer mentioned, we understand that it is crucial to exclude the effects of homogeneous reactions. In order to eliminate the possible leaching of Fe to form a homogeneous catalytic system, we performed a filtration operation during the catalytic reaction (Figure S17). However, the description of this section was not clear enough, which led to the misunderstanding of reviewers. Fe–**NDI** was filtered out of the reaction system after 4 hours of the alkylation reaction, and then the filtrate was kept in the light until 14 hours. The results showed that the yields did not change after the Fe–**NDI** was filtered out. In addition, we also performed filtration experiments in

parallel after 6 and 8 hours, respectively, and the yields did not improve further. These results indicated that Fe–**NDI** was really a stable heterogeneous catalyst and there was no FeCl₃ being released to form a homogeneous catalytic system. We have modified this statement for easy understanding by reviewers and readers in the revised manuscript. To further validate this point, the filtrate was performed on the ICP-MS test, the results showed that the Fe ions were not detected in the solution, further confirming the high stability of Fe–**NDI** during the reaction process.

Because of the reference to FeCl₃ through the LMCT process to form chlorine radicals for efficiently activating the C–H bonds process, this C–H bond activation process is consistent with the literature mentioned by the reviewers to ensure the efficient formation of alkyl radicals to participate in the subsequent transformation. However, it is worth noting that this photocatalytic system is significantly different from the homogeneous catalytic system in the literature (*Chem Catal.* 2022, 2, 1211–1222). (1) The required excitation light shifted from 390 nm to 455 nm, visible light catalysis is more meaningful in the field of photocatalysis; (2) The photocatalysis reaction time was cut in half, indicating that this system possessed a higher photocatalysis activity; (3) Fe–**NDI** as a heterogeneous catalyst avoided the introduction of free metal ions into the system reducing the difficulty of the separation; (4) There was virtually no attenuation of the reactive activity in the multiple rounds of cycling, which has more potential application value than the homogeneous catalytic system. These clear advantages reflect the feasibility of an energy transfer-mediated multiphoton synergistic excitation strategy by Fe–**NDI** integrating the Fe(III)–Cl chromophore LMCT process and **NDI** activation oxygen through the energy transfer process.

We read the literature referred to by the reviewer in comments 7, where FeCl₃ (0.057 M) in aqueous HCl solution (9 M) showed significant absorption peaks at 315 and 360 nm, which was attributed to FeCl₄[−] species. Considering whether a consistent phenomenon exists in acetonitrile solutions, we dissolved FeCl₃ in acetonitrile and performed UV-Vis absorption spectroscopy tests. Without the addition of HCl, the solution already showed distinct absorption peaks at 310 and 360 nm. This phenomenon might be due to the fact that the absorption peaks of the Fe–Cl chromophore in acetonitrile are 310 and 360 nm and were not specifically attributed to the FeCl₄[−] species. In the UV-Vis titration experiment shown in Figure 2a, where a trace amount of hydrochloric acid solution was slowly added to the acetonitrile solution containing Fe–**NDI**, the significantly enhanced absorption peaks observed at 310 and 360 nm should be attributed to the Fe–Cl chromophore on Fe–**NDI** in the acetonitrile solution. In order to rule out the possibility that the specific absorption peaks could come from the potentially homogeneous species resulting from the partial dissociation of heterogeneous catalysts, the heterogeneous Fe–**NDI** was filtered, and the UV-Vis spectra of the filtrate showed the absence of absorption peaks at 310 and 360 nm (Figure S12). In addition, the filtrate was performed the ICP-MS test, which again resulted in the absence of free Fe ions. The above experimental results indicated that the UV-Vis absorption peaks at 310 and 360 nm attributed to the LMCT absorption bands of the Fe–Cl chromophores of the

Fe–**NDI** bound to the Cl ions.

After the addition of CHCA, the absorption band of Fe–**NDI** continued to decrease in the UV-Vis spectrum above 375 nm, which was similar to the phenomenon in Figure 2a, probably due to the dilution effect of the added solution leading to the decrease of UV-Vis absorption intensity. However, no significant LMCT absorption peaks were observed during the UV-Vis titration, which might be due to the fact that the carboxyl group in CHCA had a greater steric hindrance than the Cl ion thus exhibited a weaker coordination ability. Since the UV-Vis spectra of CHCA did not illustrate our conclusions well, we have removed the relevant spectra in the revised supporting information and the relevant conclusions in the revised manuscript.

As mentioned in the literature by the reviewer (*Inorg. Chem.* 2023, **62**, 6661–6673), Fe–**NDI** was formed by the self-assembly of Fe ions with **NDI** ligands, which inherited the fluorescence emission properties of **NDI** ligands. There was a significant enhancement in the fluorescence emission of Fe–**NDI** after the addition of CHCA, while the fluorescence emission of the filtrate disappeared after separating Fe–**NDI** from the system by filtration (Figure S20). This result suggested that the fluorescence enhancement was due to the interaction of CHCA with heterogeneous Fe–**NDI**, excluding the dissociation of Fe–**NDI**.

Reviewer's Comments (2): The net reaction detailed in Fig. 1a and the object of the computational inquiry (Fig. 4, Table S5, and Fig. S17) is:

and that this reaction occurs at an iron catalyst, is at heart an example of Fenton chemistry. I encourage the authors to consult the following references:

Barton, D. H. R.; Dollar, D. *Acc. Chem. Res.* 1992, 25, 504–512.

Therein, the authors will find similar observations regarding the absence of evidence for OH*, as is commonly invoked in Fenton mechanisms, as well as an alternative mechanism for the same overall oxidation of cyclohexane. As Fenton chemistry generally involves H₂O₂ as a reagent, the authors should exclude its involvement with a H₂O₂-specific quenching agent, e.g. catalase or CuSO₄/2,9-dimethyl-1,10-phenanthroline, as DMPO and Benzoquinone do not distinguish between O₂^{*-} and H₂O₂.

Responses: We are grateful to the reviewer for the suggestion. In the mechanism validation section, we considered that the presence of multiple reactive oxygen species in the system would influence the oxidation process. Since the presence of Fe(II) may induce the Fenton reaction to cleave H₂O₂ releasing highly reactive [•]OH, we added *t*-BuOH as a characteristic inhibitor of [•]OH in cyclohexane oxidation experiments (Figure 3g). The results showed that the addition of *t*-BuOH decreased the yield of cyclohexanone by only 12%, indicating that the [•]OH produced by H₂O₂ homolytic cleavage had little effect on the oxidation reaction of cyclohexane in this reaction system. According to the reviewer's suggestion, to further confirm the extent of the effect of H₂O₂ on

cyclohexane oxidation, we added catalase (30000 U) as the H₂O₂-specific quenching agent to the cyclohexane oxidation reaction. The results showed that the reaction yield decreased by only 14%, which was consistent with that of the t-BuOH experiment, further indicating that the Fenton reaction is not a major player in the oxidation process. We have added some descriptions in the revised manuscript to describe the results and added the data in the revised Supporting Information.

Reviewer's Comments (3): There are a number of inconsistencies between what is shown in Fig. 1, Fig. 4, Table S5, Fig. S17, and the text relating to the details of the mechanism of the substrate (probably better described as RH₂ for consistency) oxidation reaction.

Panel (a) in Fig. 1 shows the oxidation of the Fe^{II} site of the catalyst by ¹O₂ to form an Fe^{III}OOH intermediate, neglecting an H* source. This is followed by the evolution of H₂O from Fe^{III}-OOH to yield the key Fe^{IV}=O species. Heterolytic cleavage of the Fe^{III}-O-OH bond after the addition of a proton would yield an Fe^V=O species, while homolytic cleavage of the Fe^{III}-O-OH bond would yield the Fe^{IV}=O species presented. Therefore, in the homolytic case presented, another source of net H* is required from the scheme to generate both the Fe^{III}OOH and the H₂O. However, the figure and text specify Cl* generated from a separate photolytic pathway is performing the hydrogen abstraction from the RH₂ substrate to yield an RH* radical. The subsequent addition of the RH* radical to the Fe^{IV}=O yields the Fe^{III}O-RH intermediate. Finally, the loss of a proton is shown to generate an Fe^{III}-O-R* radical species which disobeys charge balance. Finally, exchange with a Cl⁻ anion yields Fe^{III}-Cl and the R=O product, regenerating the catalyst. If Cl* is performing HAT to generate RH*, Cl⁻, and H⁺, there is no source of H* in Fig. 1a which is a critical oversight for at least two of the steps shown.

This critique is corroborated by the authors' own DFT calculations as shown in Fig. 4. Here the authors explicitly show the addition of an exogenous electron between intermediates S1 and S2 and the addition of an exogenous proton between steps S2 and S3. However, in this figure, the authors again neglect a net H* source somewhere between S3 and S4 to generate H₂O as well as not noting the addition of the exogenous chloride anion in steps S6 to T1. Shown in the reaction path, there is an addition of a net H* and a loss of a net H* but this still leaves the evolution of H₂O unbalanced. Ostensibly, the initial electron source could originate from the photoexcited NDI* shown at the top left of Figure 1, thus allowing the acidic solution (NH₄Cl in MeCN) to provide the proton and leaving a positive ligand to be reduced to close the catalytic cycle, yet this still doesn't account for the evolution of water.

Responses: We are grateful to the reviewer for the suggestion. Figure 1 shows a brief schematic of the entire Fe-NDI-catalysed oxidation process, ignoring the partial addition of protons and electrons. Its corresponding proton and electron originated transfer mechanism is described in detail in Figure 4.

In the content concerning the DFT calculation of the oxidation reaction mechanism (Figure

4), we apologize for the confusion caused to the reviewers due to our omission of the details in the figures. Throughout the oxidation process, the following oxidation equation is followed.

The chlorine radicals (Cl^\cdot) abstracted a hydrogen atom of the substrate RH_2 via the HAT process to produce $\cdot\text{RH}$ and HCl . Subsequently, the *in situ* generated Fe(II) node reduces $^1\text{O}_2$ to produce $\text{O}_2^{\cdot-}$ (S1 to S2). With the help of a proton and an electron, $\text{O}_2^{\cdot-}$ combines with the Fe node to produce Fe-OOH intermediate (S2 to S3). The O-O bond was stretched with the assistance of the binuclear iron node (TS1), which was later heterolytically cleaved upon the incorporation of a proton and an electron to produce the Fe(IV)=O species accompanied by the generation of H_2O (S3 to S4). And then Fe(IV)=O combined with the previously generated $\cdot\text{RH}$ to produce the Fe(III)-O-RH species (S4 to S5). The departure of two electrons and one proton allowed the generation of cyclohexanone. At the same time, the previously generated Cl^- ions recombined with Fe(III) allowing Fe-NDI to complete the catalytic cycle (S5 to T1). The proton released from the HCl produced by the HAT process together with the electrons and protons released by the S4 to T1 processes satisfied the two electrons and two protons required for the S1 to S4 processes. Overall, all the electrons and protons of the oxidation reaction process were in equilibrium, and the whole reaction system satisfied Formula. 1 with no entry of exogenous protons and electrons. We have modified Figure 1, Figure 4 and Table S5 in the revised manuscript and supporting information to ensure the consistency of the content related to DFT calculations.

Reviewer's Comments (4): There are some apparent errors in the spin states of the calculations as shown in Fig. 4. Addition of singlet O_2 and a single electron to S1 (presumably "singlet 1" despite being composed of a $d_5 \text{Fe}^{\text{III}}$ and a $d_6 \text{Fe}^{\text{II}}$) to form S2 (presumably "singlet 2") is impossible as addition of a single electron to a singlet state must result in even multiplicity, with a doublet the expected spin state. If Table S5 is to be believed, which differs from Fig. 4, then there is an error between S2 and S3 where a net hydrogen atom is added to the system which would necessarily result in an even multiplicity spin state. If S and T do not refer to singlet and triplet spin states respectively, this should be explicitly stated somewhere.

Furthermore, the fact that the Fe^{III} centers of pristine Fe-NDI are high-spin, as evidenced by EPR, complicates spin-state assignment. The Fe^{III} sites in the crystal structure are bridged by two carboxylate ligands, which are well known to behave as superexchange pathways, resulting in some antiferromagnetic coupling of the iron centers, which can be expected to at least partially quench the EPR signal. It would be helpful to quantify the (total spin/total Fe) of the samples in the EPR chamber. It could reveal either enhanced antiferromagnetic coupling upon the addition of CHCA, HCl and NH_4Cl , which would be consistent with the resistivity measurements, or the generation of low-spin Fe^{III} .

Responses: We are grateful to the reviewer for the suggestion. We are sorry for the ambiguity

created by our failure to label the meanings of the acronyms in the DFT calculation. “S” refers to the “Steady state”, while “TS” refers to the “Transition state”. We have explained it in the revised supporting information. EPR tests were considered to characterize the change of valence state of Fe nodes in Fe–**NDI** after irradiation with a 455 nm LED to further validate the occurrence of the LMCT process. But, as the reviewer said, the EPR signals of the binuclear Fe(III) species would be silenced due to the antiferromagnetic coupling effect, while the weak signals in the current EPR spectra might be caused by the test environment and the slight changes in EPR signals were not enough to support the conclusion that LMCT occurs. Mössbauer spectroscopy is an ideal and accurate method to selectively probe the geometry and purity of Fe environments in a sample. According to the reviewer’s suggestion, the zero-field ^{57}Fe Mössbauer spectrum of Fe–**NDI** tests was executed at 80 K. The results showed a symmetric doublet with isomer shift $\delta = 0.51 \text{ mm s}^{-1}$ and quadrupole splitting $|\Delta\text{EQ}| = 0.81 \text{ mm s}^{-1}$, which was attributed to the high-spin Fe(III) species in Fe–**NDI** (Figure 2f). Irradiation of the Fe–**NDI** acetonitrile suspension containing HCl with a 455 nm LED gave a mixed peak pattern, on the zero-field ^{57}Fe Mössbauer spectrum. Through data fitting, the red peak pattern shows isomer shift $\delta = 0.52 \text{ mm s}^{-1}$ and quadrupole splitting $|\Delta\text{EQ}| = 0.83 \text{ mm s}^{-1}$, which was attributed to a high-spin Fe(III) iron species. The blue peak pattern shows isomer shift $\delta = 1.32 \text{ mm s}^{-1}$ and quadrupole splitting $|\Delta\text{EQ}| = 3.10 \text{ mm s}^{-1}$, which was attributed to a high-spin Fe(II) iron species. A 27% conversion from the high-spin Fe(III) species to high-spin Fe(II) species in Fe–**NDI** under the irradiation of 455 nm LED further verifies the occurrence of the LMCT process. The formed highly electrophilic Cl radical can grab the H atom from the C–H bond of alkanes to generate alkyl radicals for participating in the subsequent functionalization process, accompanied by Fe(II) formation. The Mössbauer spectrum of Fe–**NDI** before and after the oxidation reaction and related descriptions were also added to the revised manuscript.

Reviewer’s Comments (5): The scheme reported in the SI (Table S5) is inconsistent with Fig. 4, showing instead that the S1 to S2 reaction does not involve the addition of an electron (also, in Table S5 all “ST1” should be “TS1” to be consistent with Fig. 4, Fig. S17, and the text). The authors did not provide coordinates for any of the 8 optimized structures, nor a table of relevant structural parameters, which precludes identification of reasonable parameters to assist in correcting this error (e.g., is the O–O bond length in S2 reasonable for a superoxo or a peroxide?). Moreover, by eye the Fe–O₂ bond length appears to be exceptionally long. Assuming the other Fe–O bond lengths are similar to those reported in the crystal structure parameter table, the Fe–O₂ bond must be greater than the $\sim 1.9\text{--}2.1 \text{ \AA}$ of reported Fe^{III}-superoxos. As shown in Fig. 4, the text states that a proton is added to S2 to generate S3, which again is inconsistent with table S5 which states hydrogen and electron addition occurs at this step.

Responses: We are grateful to the reviewer for the suggestion. We apologize for the discrepancy between the manuscript Figure 4 and the supporting information Table S5 related to DFT calculations due to our carelessness. The $^1\text{O}_2$ generated by ligand photosensitization during the

S1–S2 process gets an electron from Fe(II) to generate $O_2^{\cdot-}$, in which O–O bond length in S2 is 1.23 Å. With the help of a proton and an electron, $O_2^{\cdot-}$ combined with the Fe node to produce Fe–OOH intermediate (S2 to S3). We have corrected writing errors in the text and harmonized all elements of the DFT theoretical calculations. We have provided the atomic coordinates of the eight intermediates in the DFT theoretical calculations in the revised supporting information to help reviewers and readers understand and recognize the reaction intermediates.

In response to the reviewer's suggestion that the Fe– O_2 bond in the S2 intermediate state in the DFT calculation is too long, we have recheck the DFT calculation of the oxidation process. The distance between the $O_2^{\cdot-}$ and Fe node at the lowest energy state is 3.29 Å, which is more inclined to bind $O_2^{\cdot-}$ on the surface of Fe–**NDI** through physical adsorption. Figure 4 and Table S5 were revised for consistency, and the full manuscript was carefully checked to avoid writing errors impacting the reviewer's reading.

Reviewer's Comments (6): Assuming Table S5 is the most correct, it is very odd that a net hydrogen atom is added to the complex between the transition state and S4. This begs the question of how the authors define a transition state. Distortion in either direction along the imaginary mode of the transition state should lead to reactants or products, but never have I seen an example of the addition of exogenous species to a transition state to generate products. The authors don't specify how this TS was acquired, e.g. nudged elastic band is a common method on VASP, though how one would add a net hydrogen atom from space is unclear.

Responses: We are grateful to the reviewer for the suggestion. We apologize for the inadvertent discrepancy between Figure 4 and Table S5, and the lack of indication of the origin of the electrons and protons. We have corrected the errors and harmonized the content related to DFT theory calculations in the corrected manuscript and supporting information. As answered earlier in **comment 3**, Fe–**NDI** catalyzes the reaction of alkanes with oxygen to produce ketone products and water under irradiation with the 455 nm LED and the reaction system is free of exogenous hydrogen atoms. The protons for the S2–S3 process and the TS1–S4 process are derived from protons ionized by HCl after the HAT process and protons released by the S5–S6 process, respectively. In terms of the overall process, the oxidation reaction course was proton conserving.

Reviewer's Comments (7): While it is clear there is some change occurring to the Fe sites of Fe–**NDI** in the presence of Cl^- upon exposure to HCl or NH_4Cl based on the UV-Vis data (Fig. 2a), evidence of coordination of Cl^- to the Fe sites would be better presented by highlighting the emergence of the 2 strong absorption bands ($Cl-Fe^{III}$ LMCT) at ~310 and ~360 nm, and the lack thereof in the CHCA spectrum (S15). Adding an analogous supporting UV-Vis spectra for the NH_4Cl case would make this claim unambiguous if those same two LMCT bands are observed. The evolution of those two bands in the presence of HCl are very similar to those observed in solutions of $FeCl_3$ at high HCl concentrations:

Microchimica Acta, 2020, 187, 488

and high LiCl concentrations:

Chem. Geol. 2006, 231, 326–349

moreover, those two bands are diagnostic of FeCl_4^- :

J. Am. Chem. Soc. 1953, 75, 1435–1443

and have been assigned to FeCl_4^- LMCT bands; which can persist upon ligation, coincident with notable red-shift of the ligand $\pi-\pi^*$ band:

Open J. Inorg. Chem. 2013, 3, 7–13

Responses: We are grateful to the reviewer for the suggestion. We have explained in detail the change process of the UV-Vis spectrum in your first comment. Moreover, according to the reviewer's suggestion, we tried our best to attempt the titration of NH_4Cl , and no obvious Fe(III)–Cl LMCT absorption peaks were found during the titration process. We speculated that the possible reason for this was the fact that NH_4Cl was not strongly acidic compared to HCl, which made it difficult to accelerate the protonation of the solvent coordinated to the Fe node, and thus the rate of binding of Cl ions to the Fe node was slower. However, the rate of C–H activation was reduced using NH_4Cl to be able to better match the kinetics of oxygen activation by Fe–**NDI**, and this two photons excitation process promoted the efficient and highly selective oxidation of cyclohexane to cyclohexanone.

Reviewer's Comments (8): The authors need to state somewhere what excitation wavelength was used for the fluorescence experiments (Fig. 2a), if the 455 nm LED was used, it would also be beneficial to provide a spectrum showing the bandwidth of this source.

Responses: We are grateful to the reviewer for the suggestion and we are very sorry for the negligence. The excitation wavelength was 455 nm for the fluorescence experiments and it has been added in the revised manuscript. All experiments were performed under the illumination of a 455 nm LED with a half peak width of 30 nm (Figure S11). We have added some words to describe this point in the revised supporting information.

Reviewer's Comments (9): In the Table 1 caption for conditions d, “CHCA” should be “4”, and in the description, line 265, page 11, “5a” should be bold.

Responses: We are grateful to the reviewer for the careful suggestion and sorry for the careless negligence. We have corrected the content mentioned by the reviewers in the revised manuscript and carefully checked the full manuscript and supporting information to ensure the accuracy of the description to improve the quality of our publication.

Reviewer's Comments (10): Unless the EPR and XPS spectra were collected during in-situ irradiation (if so, this should be explicitly stated, and dark control conditions should be presented), the EPR and XPS findings to support partial reduction of the system to generate Fe^{II} is

unconvincing. The appearance of the Fe–**NDI** + HCl/CHCA peaks assigned as Fe^{II} 2p_{1/2} coincides with a drastic ~+3 eV shift of the assigned Fe^{III} 2p_{1/2} peaks. While there is clearly an emergence of additional features in the XPS spectra, it is far more likely attributable to multiple Fe^{III} environments due to coordination of Cl⁻ or cyclohexanecarboxylate. Moreover, the absence of an increased absorbance peak for the CHCA soaked sample near 455 nm (Fig. S15) makes it unlikely that LMCT from coordinated CHCA is resulting in the reduction of Fe^{III}. The diminished EPR signal could just as easily be assigned to a conversion from high-spin to low-spin Fe^{III} due to a change in coordination environment or as a consequence of enhanced antiferromagnetic coupling as mentioned above. In the absence of the higher-field of the EPR data, the reduction of Fe^{III} cannot be concluded from this data, only a lower concentration of high-spin Fe^{III}.

Responses: We are grateful to the reviewer for the suggestion. As the reviewer mentioned, the binuclear Fe(III) sites in the crystal structure were bridged by two carboxylate ligands, which were well known to behave as superexchange pathways, resulting in some antiferromagnetic coupling of the iron centers, which can be expected to at least partially quench the EPR signal. The weak signals in the current EPR spectra might be caused by the test environment by the test environment and the slight changes in EPR signals were not enough to support the conclusion that LMCT occurs.

Mössbauer spectroscopy is an ideal method to selectively probe the geometry and purity of Fe environments in a sample. As the reviewer's suggestion, we supplemented the Moessbauer spectroscopy. The zero-field ⁵⁷Fe Mössbauer spectrum of Fe–**NDI** tests was executed at 80 K. The results showed a symmetric doublet with isomer shift $\delta = 0.51 \text{ mm s}^{-1}$ and quadrupole splitting $|\Delta EQ| = 0.81 \text{ mm s}^{-1}$, which was attributed to the high-spin Fe(III) species in Fe–**NDI** (Figure 2f). Irradiation of the Fe–**NDI** acetonitrile suspension containing HCl with a 455 nm LED gave a mixed peak pattern, on the zero-field ⁵⁷Fe Mössbauer spectrum. Through data fitting, the red peak pattern shows isomer shift $\delta = 0.52 \text{ mm s}^{-1}$ and quadrupole splitting $|\Delta EQ| = 0.83 \text{ mm s}^{-1}$, which was attributed to a high-spin Fe(III) iron species. The blue peak pattern shows isomer shift $\delta = 1.32 \text{ mm s}^{-1}$ and quadrupole splitting $|\Delta EQ| = 3.10 \text{ mm s}^{-1}$, which was attributed to a high-spin Fe(II) iron species. A 27% conversion from the high-spin Fe(III) species to high-spin Fe(II) species in Fe–**NDI** under the irradiation of a 455 nm LED further verifies the occurrence of the LMCT process. The results of Mössbauer spectroscopy are consistent with the results of the XPS test, showing the valence state transformation of Fe nodes in Fe–**NDI**, which further proves the occurrence of the LMCT process. The Mössbauer spectrum and related descriptions have been added to the revised manuscript.

Reviewer's Comments (11): On page 7, line 175, the authors suggest the decreased resistance is a consequence of Cl⁻ coordination; however, the lowest resistance was achieved with the chloride-free acid CHCA. Thus, it appears more likely that ligation change as a consequence of acid exposure is the most logical origin of decreased resistance.

Responses: We are grateful to the reviewer for the suggestion and we agree with the reviewer's

viewpoint. We only compared the improvement in the photocurrent test of Fe–**NDI** and the decrease in resistance compared to the initial state after the addition of HCl, NH₄Cl, and CHCA, respectively, which suggests that the electron transfer ability of Fe–**NDI** was significantly improved after coordination. We did not compare the effect of adding different additives on the photoelectric properties of Fe–**NDI**. However, Fe–**NDI** showed the lowest resistance when CHCA was added, which may be due to the ligation change mentioned by the reviewer.

Reviewer’s Comments (12): On page 8, lines 204–210 are very confusing; the authors state that after removing the heterogenous photocatalyst, “the filtrate continued to react for up to 14 hours under standard reaction conditions”, but the catalytic yield is unchanged, and Fig. S13 shows that the yield at 4, 8, and 16 hours as unchanged. This sentence should be revised. Either i) the reaction stops after removal of the photocatalyst, ii) a change occurs as a result of imperfect filtration, or iii) the active photocatalyst is homogeneous.

Responses: We are grateful to the reviewer for the suggestion and sorry for this ambiguous statement, which led to the misunderstanding of reviewers. Fe–**NDI** was filtered out of the reaction system after 4 hours of the alkylation reaction, and then the filtrate was kept in the light until 14 hours (Figure S17). The results showed that the yields did not change after the Fe–**NDI** was filtered out. In addition, we also performed filtration experiments in parallel after 6 and 8 hours, respectively, and the yields did not improve further. These results indicated that Fe–**NDI** was really a stable heterogeneous catalyst and there was no FeCl₃ being released to form a homogeneous catalytic system. We have modified this statement for easy understanding by reviewers and readers in the revised manuscript.

Reviewer’s Comments (13): On line 288, the authors suggest both ¹O₂ and O₂^{*-} were observed from radical trapping agents, but DMPO is promiscuous and O₂^{*-} and O₂²⁻ (H₂O₂) are indistinguishable. Catalase is commonly used as an H₂O₂-selective quencher in similar cases across a range of solvents including MeCN, and can tolerate mildly acidic conditions.

Responses: We are grateful to the reviewer for the suggestion. According to the reviewer's suggestion, we supplemented the EPR test with the addition of catalase. However, due to the limited solubility of catalase in acetonitrile solution, catalase did not quench H₂O₂ efficiently during the EPR test and the results were the same as before. According to relative reference (*J. Am. Chem. Soc.* 2019, **141**, 19110–19117), the signal of [•]OH can be specifically quenched by methanol in the EPR test. Therefore, we used methanol as a solvent to undergo the EPR test under irradiation with 455 nm LED after adding DMPO (Figure 3f). The result showed the characteristic signal peaks of standard O₂^{*-} (*J. Am. Chem. Soc.* 2022, **144**, 18586–18594), which indicated that Fe–**NDI** could activate oxygen to form O₂^{*-} through electron transfer. Despite the possibility of generating other reactive oxygen species (O₂²⁻) during the reaction, in combination with free radical quenching experiments, it can be demonstrated that the main reactive oxygen species involved in

the oxidation reaction are the singlet oxygen and superoxide radicals. The revised EPR spectra and associated descriptions have been added to the revised manuscript.

Reviewer's Comments (14): The acronym CHCA needs to be defined earlier in the manuscript text or in the caption of Fig. 2 and Table 1. CHCA is not defined until page 11, while it is referenced in all but one panel of Fig. 2 and Table 1 currently on pages 5 and 9, respectively. The first time a panel that includes CHCA is discussed in the text is on line 100 (page 4), and without that definition, it is difficult to contextualize the findings.

Responses: Thanks very much to the reviewer for the suggestion and sorry for this careless negligence. We have defined the acronym CHCA in the caption of Fig. 2 to be easily understood by reviewers and readers in the revised manuscript.

Reviewer's Comments (15): Caption of Fig. S7 should change “The first weightlessness could belong to the loss of free and coordination solvent. The second weightlessness was attributed to the decomposition of Fe–NDI.” to “The first weight loss event could be due to the loss of free and coordinated solvent. The second weight loss event was attributed to the decomposition of Fe–NDI.”

Responses: We are grateful to the reviewer for the attentive suggestions. According to the reviewer's suggestion, we have revised the corresponding description in the revised manuscript. The reviewer's suggestions help us improve the English of the manuscript. Meanwhile, we also examined the remaining parts of the full text for further improving the quality of the article.

Reviewer's Comments (16): Figure S11 x-axis label should be “amount” instead of “mount”.

Responses: Thanks to the reviewer for the careful suggestions. We have modified “mount” into “amount” in Figure S11 x-axis label. Moreover, we have carefully checked and done our best to revise the full manuscript and supporting information to improve the quality of our publication.

Reviewer's Comments (17): Figures S15 and S16 captions need to be more detailed w.r.t. concentrations of species other than CHCA, and Fig. S16 caption needs to include excitation wavelength.

Responses: Many thanks to the reviewer for the suggestions and sorry for this careless negligence. According to the reviewer's suggestion, the related details of the experiments have been added to the revised supporting information. At the same time, we carefully checked and did our best to refine all the details of the data as much as possible.

Reviewers' Comments:

Reviewer #1:

Remarks to the Author:

The revision has significantly improved the quality of the manuscript. In particular, the Moessbauer experiments unambiguously provide evidence in support of the generation of Fe(II) center by Fe(III)-Cl bond homolysis. The work is well done and can be suitable for publication in Nature Communication, subject to revision.

a) the mechanism of the C-C coupling step in presence of electron withdrawing alkenes should be shown as a scheme. Also the authors use dicyano styrene as the radical trap and not styrene as mentioned in many places. Show the SET step in the scheme.

b) Same is true for the decarboxylation mechanism. A mechanism either in the text or in the SI would be helpful

c) I am not sure how product quantification and the yields are performed. Is it done by gas chromatography? please include the relevant data in the SI. I could only find ESI-MS data, which do not say anything about the purity or yield.

d) evidence of the generation of Fe(IV)=O is important. the authors claimed to have obtained a Fe=O vibration in the rRaman. Please support this with $^{18}\text{O}_2$ labeling experiments. Alternatively, can Moessbauer be performed to show the generation of Fe(IV)=O moiety.

e) I agree with the authors that an Fe(IV)=O center may react with Alkyl radicals to form the Fe(III)-alkoxides. I do not understand why the alkoxides perform the dehydrogenation step to yield the ketone. Iron(III) centers are generally stable and no dehydrogenation generally works. This is the major problem in biomimetic catalysis as all reactions stop in the Fe(III) oxidation state and no catalysis is obtained.

f) the relevance to sMMO is unnecessary. sMMO performs a dinuclear O_2 activation and the mechanism is significantly different than in the present case. Also enzymes react with triplet oxygen.

Reviewer #2:

Remarks to the Author:

The paper has substantially been enhanced after the first revision. The issues raised in the first review have been positively responded to and corrected, and some ambiguous spectra and descriptions have been positively optimized. In my opinion, the contribution to the related state of the art is consistent and appreciable. The revisions implemented by the authors have further improved the quality of the document.

Reviewer #3:

Remarks to the Author:

The authors have satisfactorily addressed all the comments from the reviewer and the manuscript can now be accepted in its current form.

Reviewer #1:

The revision has significantly improved the quality of the manuscript. In particular, the Moessbauer experiments unambiguously provide evidence in support of the generation of Fe(II) center by Fe^{III}-Cl bond homolysis. The work is well done and can be suitable for publication in Nature Communication, subject to revision.

Response: Many thanks to the reviewer for the positive and kind comments. We will try our best to improve the quality of our publications.

a) the mechanism of the C-C coupling step in presence of electron withdrawing alkenes should be shown as a scheme. Also the authors use dicyano styrene as the radical trap and not styrene as mentioned in many places. Show the SET step in the scheme.

Responses: Thanks very much for the reviewer's careful suggestions. The scheme of the alkylation reaction mechanism have been added in the revised supporting information, which show the SET step (Figure S18).

b) Same is true for the decarboxylation mechanism. A mechanism either in the text or in the SI would be helpful.

Responses: Thanks very much for the reviewer's careful suggestions. The scheme of the decarboxylation mechanism have also been added in the revised supporting information (Figure S23).

c) I am not sure how product quantification and the yields are performed. Is it done by gas chromatography? please include the relevant data in the SI. I could only find ESI-MS data, which do not say anything about the purity or yield.

Responses: Many thanks to the reviewer for the kind suggestions. The products of alkylation and decarboxylation reactions were separated on silica gel column chromatography (EtOAc/petroleum ether) to obtain the isolated yields. Due to the low boiling points of some oxidation products, it is difficult to separate and collect them by column chromatography. Therefore, the yields of the oxidation products were determined by gas chromatography (GC) using 1,3,5-trimethoxybenzene as the internal standard. In an analytical sample containing an equimolar amount of 1,3,5-trimethoxybenzene to the substrates, the yields of the target products were calculated according to the area of the chromatographic peak. The method of determining yields of the oxidation reaction products have been added in detail to the revised supporting information.

The ESI-MS data mentioned by the reviewer corresponds to the radical trapping experiments, which is used to explore the reactive species and reaction intermediates during the reaction.

d) evidence of the generation of Fe^{IV}=O is important. the authors claimed to have obtained a Fe=O vibration in the rRaman. Please support this with ¹⁸O₂ labeling experiments. Alternatively, can Moessbauer be performed to show the generation of Fe(IV)=O moiety.

Responses: Thanks very much for the reviewer's careful suggestion. At the suggestion of the reviewer, in order to further provide the evidence of the generation of the Fe(IV)=O, the $^{18}\text{O}_2$ labelling Raman experiment was supplemented (Figure S24). The result showed a new distinct scattering peak at 789 cm^{-1} , which was reasonably attributed to Fe(IV)= ^{18}O and the observed $\Delta^{18}\text{O}/^{16}\text{O}$ of 31 cm^{-1} was in excellent agreement with related literature (*J. Am. Chem. Soc.* **2005**, *127*, 12494–12495). We have added the relevant descriptions and spectra of ^{18}O isotope labelled Raman experiments in the revised manuscript and supporting information.

e) I agree with the authors that an Fe^{IV}=O center may react with Alkyl radicals to form the Fe(III)-alkoxides. I do not understand why the alkoxides perform the dehydrogenation step to yield the ketone. Iron(III) centers are generally stable and no dehydrogenation generally works. This is the major problem in biomimetic catalysis as all reactions stop in the Fe(III) oxidation state and no catalysis is obtained.

Responses: Thanks very much for the reviewer's useful suggestion and sorry for this unthoughtful statement. At the suggestion of the reviewer, we reviewed the relevant literatures and rechecked the possible oxidation reaction pathway of the DFT calculations. According to the relevant literature (*Angew. Chem. Int. Ed.* **2019**, *58*, 8484–8488), after the formation of the Fe–OOH intermediate, the O–O bond was stretched and cleaved to produce the Fe(IV)=O species (Figure 4 TS1 to S4), accompanied by the oxidation of another Fe(III) to Fe(IV) in the dinuclear iron node. Subsequently, Fe(IV)=O species received the alkyl radical produced after the HAT process to generate the Fe(III)–O–RH intermediate (S4 to S5), which was oxidised by another Fe(IV) with the simultaneous proton departure to produce the Fe(III)–O–R* intermediate (S5 to S6), and final ketone production. The possible oxidation reaction pathway of the DFT calculations have been updated in the revised manuscript and supporting information.

f) the relevance to sMMO is unnecessary. sMMO performs a dinuclear O₂ activation and the mechanism is significantly different than in the present case. Also enzymes react with triplet oxygen.

Responses: Many thanks to the reviewer for thoughtful suggestions. In fact, inspired by the active sites of natural oxidases initially, we designed a metal–organic framework Fe–**NDI** with suitably distant binuclear iron nodes and photosensitive dye **NDI** ligands. Through a multiphoton excitation process, chlorine radicals (Cl[•]) were generated via LMCT between Fe(III) and chlorine ions to activate the C–H bonds via the HAT process, and the **NDI** activated oxygen to generate $^1\text{O}_2$. Subsequently, the $^1\text{O}_2$ was combined by binuclear iron node to generate a high-valent Fe(IV)=O species, which completed C–H bond oxidation in combination with alkyl radicals. As said by the reviewer, this oxidation process catalyzed by Fe–**NDI** was distinct from the oxidase reacting with the triplet oxygen. After careful consideration, we agree with the reviewer's suggestion and modified the relevant description about sMMO in the revised manuscript.

Reviewer #2:

The paper has substantially been enhanced after the first revision. The issues raised in the first review have been positively responded to and corrected, and some ambiguous spectra and descriptions have been positively optimized. In my opinion, the contribution to the related state of the art is consistent and appreciable. The revisions implemented by the authors have further improved the quality of the document.

Responses: Many thanks to the reviewer for the positive assessment of our work. We will try our best to improve the quality of our publications.

Reviewer #3:

The authors have satisfactorily addressed all the comments from the reviewer and the manuscript can now be accepted in its current form.

Responses: Many thanks to the reviewer for the positive assessment of our work. We will try our best to improve the quality of our publications.

Reviewer #1

(Remarks to the Author)

The manuscript can be accepted for publication. I need two minor alterations

a) Figure S24: Either the experiment should be tried again or the ^{18}O data should be removed. The 789 cm^{-1} peak in no way represents a real vibrational band as the line width is too small and is not consistent with the ^{16}O peaks that are vanishing. I suppose that under the low pressure of $^{18}\text{O}_2$ used the $\text{FeIV}=\text{O}$ species is not generated at all in the $^{18}\text{O}_2$ experiment. Please note that the shape and line width of the band disappearing and the band appearing in the rRaman data should be same.

b) In Figure 4: Typo Cl^+

Reviewer #1:

The manuscript can be accepted for publication. I need two minor alterations

a) Figure S24: Either the experiment should be tried again or the ^{18}O data should be removed. The 789 cm^{-1} peak in no way represents a real vibrational band as the line width is too small and is not consistent with the ^{16}O peaks that are vanishing. I suppose that under the low pressure of $^{18}\text{O}_2$ used the $\text{Fe}^{\text{IV}}=\text{O}$ species is not generated at all in the $^{18}\text{O}_2$ experiment. Please note that the shape and line width of the band disappearing and the band appearing in the rRaman data should be same.

Responses: Thanks very much for the reviewer's useful suggestion. According to the reviewers' suggestion, the Raman test for $^{18}\text{O}_2$ was repeated several times, but unfortunately did not get the desired data. Just as the reviewer said, it is very possible that the $\text{Fe}^{\text{IV}}=\text{O}$ species is not generated at all in the $^{18}\text{O}_2$ experiment under the low pressure of $^{18}\text{O}_2$ used. In order to avoid misunderstanding by readers, we agree with the reviewer's suggestion to delete this data and the relevant statement in the revised manuscript and supporting information.

b) In Figure 4: Typo Cl^+

Responses: Many thanks to the reviewer for the careful suggestion and sorry for the careless negligence. The writing error Cl^+ in Figure 4 has been corrected to Cl^- in the revised manuscript. We have carefully checked the full manuscript again to ensure the accuracy of the description for improving the quality of our publication.